# CODEPLAN: UNLOCKING REASONING POTENTIAL IN LLMS BY SCALING CODE-FORM PLANNING

**Jiaxin Wen**[1,2]*, **Jian Guan**[2]*, **Hongning Wang**[1], **Wei Wu**[2]†, **Minlie Huang**[1]†
[1]Tsinghua University [2]Ant Group
`{jiaxinwenthu, jianguanthu, wang.hongn}@gmail.com`
`wuwei19850318@gmail.com, aihuang@tsinghua.edu.cn`

## ABSTRACT

Despite the remarkable success of large language models (LLMs) on traditional natural language processing tasks, their planning ability remains a critical bottleneck in tackling complex multi-step reasoning tasks. Existing approaches mainly rely on prompting or task-specific fine-tuning, often suffering from poor robustness and cross-task generalization. To address the limitation, we introduce CODEPLAN, a scalable framework that empowers LLMs to generate and follow *code-form plans*—pseudocode that outlines high-level, structured reasoning processes. By leveraging the structured and versatile nature of code, CODEPLAN effectively captures the rich semantics and control flows inherent to sophisticated reasoning tasks. Importantly, CODEPLAN allows automatic extraction of code-form plans from massive, wide-ranging text corpora without the need for curated, task-specific datasets. This enables it to scale up efficiently and improve LLM's reasoning capabilities across diverse scenarios. To train CODEPLAN, we construct a large-scale dataset of 2M examples that integrate code-form plans with standard prompt-response pairs from existing corpora. With minimal computation overhead during both training and inference, CODEPLAN achieves a 25.1% relative improvement compared with directly generating responses, averaged across 13 challenging multi-step reasoning benchmarks, spanning mathematical reasoning, symbolic reasoning, instruction-following, multi-hop QA, and decision-making tasks. Further analysis reveals CODEPLAN's increasing performance gains on more complex reasoning tasks, as well as significant data efficiency thanks to its generalization ability.

## 1 INTRODUCTION

With the rapid progress in pre-training and post-training (Brown, 2020; Chung et al., 2024), large language models (LLMs) have exhibited remarkable performance across a wide range of natural language processing (NLP) tasks. However, as LLMs are tasked with solving increasingly complex problems that require multi-step reasoning, such as mathematical problems (Hendrycks et al., 2021), multi-hop question-answering (Trivedi et al., 2022), and complex decision-making (Shridhar et al., 2020; Xie et al., 2024), their limited planning capability has become a critical bottleneck (Yang et al., 2023). As illustrated in Figure 1, LLMs often exhibit various failure modes in multi-step reasoning, such as repetitive steps, incoherent logic, focus drift, and early answering (Yao et al., 2022; Lanham et al., 2023). Effective planning, i.e., generating a high-level abstraction in advance (Yang, 2012; Russell & Norvig, 2016), can frame the subsequent reasoning procedure, thereby guiding LLMs through the intricate low-level steps to ultimately solve the tasks (Wang et al., 2023).

Delving deeper, LLMs' planning deficiencies largely stem from the fact that massive pre-training text corpora often do not explicitly exhibit the underlying reasoning structures, thereby obscuring the latent, high-level planning signals that LLMs should learn (Zelikman et al., 2024). To remedy this challenge, current approaches mainly frame LLMs' reasoning procedures through either advanced prompting techniques (Wei et al., 2022a; Yao et al., 2024) or task-specific fine-tuning (Zelikman et al., 2022; Guan et al., 2024). However, prompting approaches typically impose strict requirements on

---

*Equal Contribution
†Corresponding Authors

Table 1: Comparison between CODEPLAN and representative methods in LM planning, evaluated from two perspectives: (1) **Expressiveness,** including *structuring* for representing complex logic, *versatility* for diverse domains and *interpretability*; and (2) **Learning**, including *data abundance*, and training/inference *efficiency*.

| Method | Plan | Expressivness | | | Learning | |
|---|---|---|---|---|---|---|
| | | Structuring | Versatility | Interpretability | Data Abundance | Efficiency |
| **CoT** (Wei et al., 2022a) | Steps Intertwined with Surface Realization | ✗ | ✓ | ✓ | ✗ | N/A |
| **Plan-and-Solve** Wang et al. (2023) | Free-Form Natural Language Text | ✗ | ✓ | ✓ | ✗ | ✗ |
| **AMOR** (Guan et al., 2024) | Expert-Designed Finite State Machine | ✓ | ✗ | ✓ | ✗ | ✗ |
| **Predicted-PA** (Cornille et al., 2024) | Learnable Latent Codes | ✗ | ✓ | ✗ | ✓ | ✗ |
| **Quiet-STaR** (Zelikman et al., 2024) | Learnable Latent Verbal Words | ✗ | ✓ | ✗ | ✓ | ✗ |
| **CODEPLAN** (This Work) | Free-Form Programming Language Code | ✓ | ✓ | ✓ | ✓ | ✓ |

models' inherent capabilities as well as carefully designed prompts (Anagnostidis & Bulian, 2024), while task-specific fine-tuning approaches limit the models' ability to generalize to new domains.

To surmount the aforementioned issues, we propose CODEPLAN, a novel, scalable framework that empowers LLMs to generate and follow *code-form plans*—pseudocode that serves as high-level, structured blueprints of the reasoning process. By leveraging the structured and versatile nature of code (Madaan et al., 2022), CODEPLAN effectively captures the rich semantics and control flows inherent to sophisticated reasoning. As exemplified in Figure 1, code naturally supports various common reasoning structures, including the hierarchical composition of multiple subtasks (function making and calling), iterative steps (`for`-loops), and conditional multi-branch steps (`if`-statements). Importantly, CODEPLAN allows the automatic extraction of code-form plans from massive, wide-ranging text corpora that naturally embed the planning signals, bypassing the need for curated, task-specific datasets. This enables CODEPLAN to scale efficiently and improve reasoning capabilities across diverse tasks beyond specific ones like mathematical reasoning (Yu et al., 2023). Table 1 summarizes the advantages of CODEPLAN against prior work.

To train CODEPLAN, we construct a large-scale dataset with 2M examples in the form of ⟨prompt, code-form plan, response⟩. We validate the effectiveness of CODEPLAN in multiple models, including Mistral (Jiang et al., 2023) and Llama (Touvron et al., 2023; Dubey et al., 2024). Extensive experiments show that CODEPLAN consistently and significantly outperforms directly generating responses without planning, yielding a relative 25.1% performance gain averaged across 13 challenging reasoning benchmarks spanning mathematical reasoning, symbolic reasoning, instruction-following, multi-hop question answering, and decision-making tasks. These results provide compelling evidence for the models' enhanced ability to tackle complex multi-step reasoning problems. By leveraging code-form plans as an intermediate representation during training, we pioneer a scalable framework for endowing LLMs with structured, versatile, and interpretable reasoning – a capability that has remained elusive when relying solely on natural language.

In summary, this work makes several pivotal contributions:

**I.** We introduce CODEPLAN, a novel, scalable framework that empowers LLMs to generate and follow *code-form* plans—pseudocode that outlines high-level, structured reasoning processes. This framework unlocks new frontiers for structured reasoning with LLMs, transcending the limitations imposed by the obscured implicit planning signals in natural language text.

**II.** CODEPLAN allows efficient and cost-effective training data construction from massive, wide-ranging corpora, enabling promising data scalability. We exemplify this by curating a large-scale dataset comprised of 2M prompt-response pairs along with their corresponding code-form plans. This dataset also establishes a rich resource for future research on reasoning in LLMs.

**III.** We demonstrate CODEPLAN's remarkable efficacy and generality across 13 challenging reasoning benchmarks on multiple backbone models, scaling from 7B to 13B. Further analysis reveals its growing advantage over baselines as problem complexity increases, and its strong data efficiency.

## 2 METHODOLOGY

We formally define the multi-step reasoning task as follows: Given a prompt $X$ that poses a problem, the goal is to generate a response $Y$ that requires a comprehensive solution through a sequence of log-

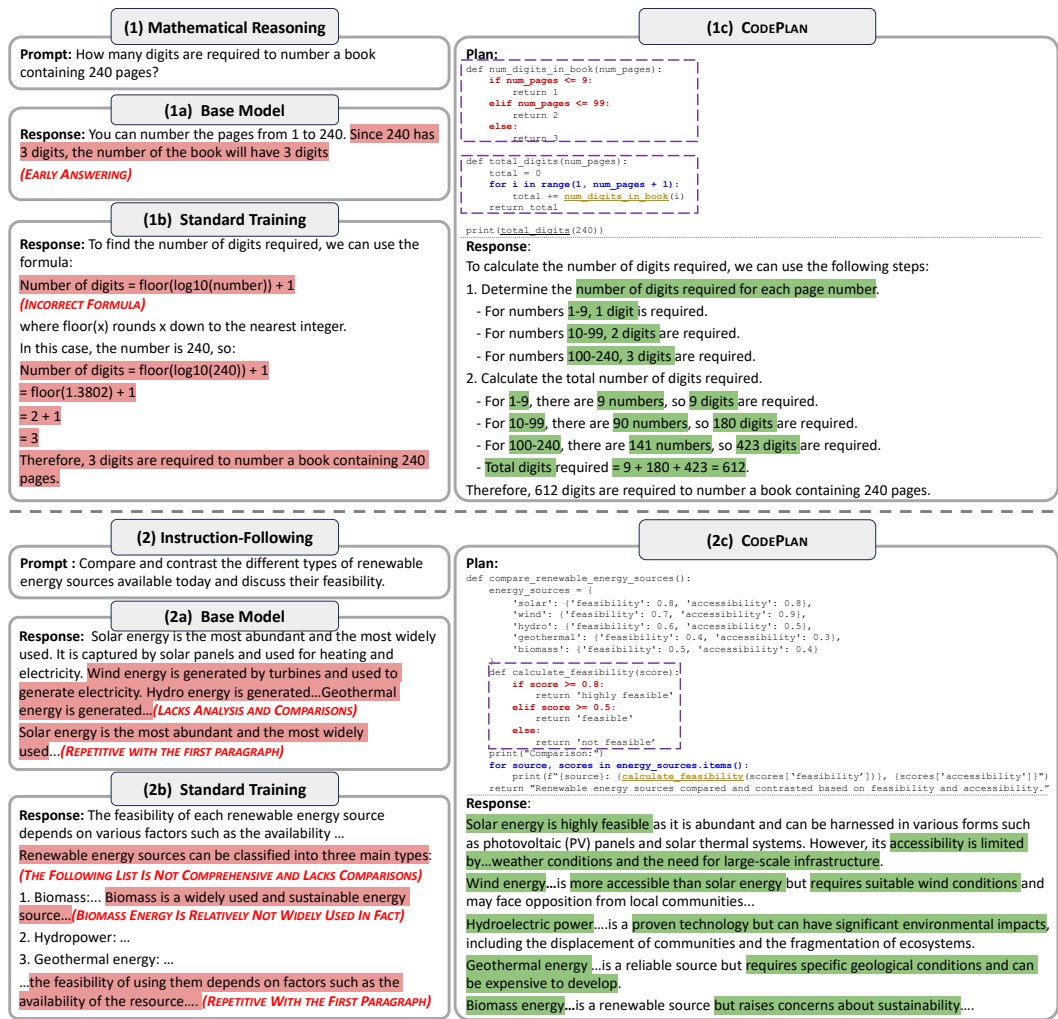

Figure 1: Two examples for the mathematical reasoning task **(Top)** and instruction-following task **(Bottom)** with Mistral-7B as the base model. **Words highlighted in red:** Unreasonable reasoning steps; **Maroon words:** Conditional branches in the plan; **Blue words:** Iterative loops in the plan; **Purple boxes:** Function making in the plan; **Golden underlined words:** Function calling in the plan; **Words highlighted in green:** Essential reasoning steps in the response adhering to the plan.

ical reasoning steps. In this section, we elaborate on the method for solving the task by decomposing the generation process (§2.1) into two stages: planning (§2.2) and surface realization (§2.3).

## 2.1 FORMALIZATION

Typically, LLMs are trained to minimize the negative log-likelihood of ground-truth outputs:

$$\mathcal{L} = -\log p(Y|X). \tag{1}$$

To imbue LLMs with structured and systematic reasoning capabilities, we propose decomposing the generation process into two stages: planning and surface realization (Reiter & Dale, 2000). The planning stage aims to generate a plan $Z$ that outlines the control flow for solving the problem $X$, while the surface realization stage then translates this plan into the final natural language response $Y$, fleshing out the low-level reasoning details. The optimization objective $\mathcal{L}$ is then modeled as follows:

$$\mathcal{L} = -\log p(Y|X) = -\log \mathbb{E}_{p(Z|X)} p(Y|X, Z), \tag{2}$$

where $p(Z|X)$ and $p(Y|X, Z)$ refer to the planning model and surface realization model, respectively. Nevertheless, marginalizing over the latent variable $Z$ is generally intractable, as the search space

could be vast. To circumvent the challenge, we minimize a variational upper bound of the loss with a posterior $q(Z|X, Y)$ (Kingma, 2013). Hence, the objective can be formulated as:

$$\mathcal{L} = -\log \mathbb{E}_{p(Z|X)} p(Y|X, Z) \leqslant -\mathbb{E}_{q(Z|X,Y)} \log p(Y|X, Z) + \text{KL}\left(q(Z|X,Y)||p(Z|X)\right).).$$

(3)

In contrast to existing approaches (Wei et al., 2022a) that intertwine planning and realization steps, the formulation explicitly disentangles these two stages, allowing for systematic generation of structured plans that effectively guide low-level reasoning steps. Moreover, it overcomes the limitations caused by the obscured planning signals in natural language data.

A well-defined $q(Z|X, Y)$ is crucial for effectively optimizing Eq. 3. We provide two principled approaches to model this posterior distribution:

**(1) Explicit Plans.** A straightforward assumption is that there exists a "gold annotator" who excels at summarizing a high-quality plan $Z^*$ for any given $(X, Y)$ pair. Under the assumption, $q(Z|X, Y)$ follows a Dirac distribution: $q(Z|X, Y) = \delta(Z = Z^*)$.

**(2) Implicit Plans.** Alternatively, plans can be modeled as learnable latent vectors that implicitly encode expert knowledge, styles, or other nuances shaping the reasoning process (Kingma, 2013). In this context, $q(Z|X, Y)$ can be defined as continuous Gaussian distribution (Kingma, 2013), discrete multinomial distribution over a learnable codebook (Van Den Oord et al., 2017; Cornille et al., 2024) or a pre-defined vocabulary (Zelikman et al., 2024). Compared to explicit representations, latent vectors might capture more subtle planning aspects. However, this approach often introduces computation overhead and may suffer from posterior collapse (Bowman et al., 2016).

In this work, we adopt the simple and explainable "Explicit Plans" approach by setting $q(Z|X, Y) = \delta(Z = Z^*)$, where $Z^*$ denotes the plan provided by a "gold annotator." We reserve the exploration of the implicit approach for future work. This setting then yields the following upper bound for Eq. 3:

$$\tilde{\mathcal{L}} = -\log p(Y|X, Z^*) - \log p(Z^*|X).$$

(4)

For simplicity, we unify the planning model $p(Z^*|X)$ and the surface realization model $p(Y|X, Z^*)$ into a single LM architecture with shared parameters $\theta$. Crucially, this formulation implies that an optimal plan $Z^*$ should achieve a delicate balance between informativeness for effective reasoning guidance and conciseness for efficient generation. This balance minimizes the combined difficulty of plan generation and subsequent realization, thereby optimizing overall model performance.

## 2.2 PLANNING

Given a prompt $X$, $Z^*$ should capture the rich semantics and control flows inherent in the reasoning path $Y$. This motivates us to use programming languages—which are Turing complete (Boyer & Moore, 1983)—as a general representation of $Z^*$, thereby framing planning as code generation.

As shown in Figure 1, this code-form representation is versatile across diverse scenarios with several compelling advantages: **(1)** It seamlessly incorporates conditional branching (i.e., `if`-statements) to dynamically adapt reasoning steps to intermediate results or contexts. **(2)** It integrates iterative loops (i.e., `for`-loops) to handle sequential data or perform repeated operations. **(3)** It defines and composes modular tools (i.e., Python functions), enhancing LLMs' abilities to craft and use tools (Cai et al., 2024). This also potentially endows our approach with the ability to build agents that can interact with external environments through specific APIs (Hausknecht et al., 2020), which is left for future work. **(4)** From a high-level perspective, it naturally supports a hierarchical reasoning structure, by defining variables and attributes upfront, addressing sub-tasks via specific functions, and orchestrating the main procedure using rigorous formal logic. This fosters the breakdown of complex reasoning problems into modular sub-components, facilitating the effective transfer of planning knowledge and enhancing the model's systematic reasoning abilities (Yang et al., 2023).

Given a dataset of $(X, Y)$ pairs, we obtain the annotation of $Z^*$ by prompting an LLM that has been extensively pre-trained on code data, as detailed in §3.1. Specifically, we instruct the model to generate a Python-style pseudocode that outlines the reasoning structure for solving the problem $X$ and arriving at the response $Y$. We do not mandate the generated plans to be executable in light of three considerations: **(1)** encoding reasoning logic using code is sufficient for generality and scalability, and execution might be unnecessary; **(2)** generating pseudocode is more token-efficient than executable code; and **(3)** our pilot study finds that existing LLMs still struggle to generate fully

Table 2: Instruction for generating the code-form plan for a given prompt-response pair.

```
Prompt: {{Prompt}}
Response: {{Response}}

Given a prompt-response pair, your task is to describe the high-level logic of the response
using a pseudo Python code.  Such that following this code, models can easily generate the
response.

The code should balance conciseness and informativeness.
The code should be high-level, instead of replicating low-level details in the response.
The code should be less than 200 words (adjust its length based on response lengths).
```

executable code plans for various tasks, as this requires a comprehensive understanding of various libraries, APIs, and domain-specific knowledge. By circumventing the execution step, we can focus on the core challenge of generating structured plans that capture the underlying reasoning logic, without additional complexities. After obtaining the dataset of $(X, Z^*, Y)$ triples, we optimize the model for plan generation by minimizing the second term of Eq. 4 (i.e., $-\log p(Z^*|X)$).

## 2.3 SURFACE REALIZATION

Next, we proceed to the surface realization stage, aiming to generate a comprehensive response $Y$ to the prompt $X$ under the guidance of the high-level code-form plan $Z^*$. To this end, we optimize the model by minimizing the first term of Eq. 4 (i.e., $-\log p(Y|X, Z^*)$).

As illustrated in Figure 1, this enables the controllable generation of $Y$ that adheres to the human-readable plan $Z$. For instance, when realizing `if`-statements, the model is explicitly conditioned on the multi-branch conditions specified in the code, ensuring adherence to the intended logic. Similarly, when expanding `for`-loops, the model is guided to systematically process each element, following the encoded iteration logic, which is critical for tasks involving iterative reasoning or sequential decision-making (Shridhar et al., 2020; Xie et al., 2024). This tight coupling between the code-form plan and the final reasoning path enables the model to produce coherent, logically sound solutions that faithfully reflect the intricate reasoning structure.

## 3 EXPERIMENTS

### 3.1 TRAINING DATA CURATION

To facilitate effective learning of planning and surface realization, we curate a large-scale training dataset of examples in the form $(X, Z^*, Y)$. We start from WebInstruct (Yue et al., 2024) that is automatically constructed from raw web data and contains diverse prompt-response pairs, and prompt Llama-3-8B-Instruct (Dubey et al., 2024) to synthesize $Z^*$. Table 2 shows the prompt. We use nucleus sampling (Holtzman et al., 2020) ($p = 0.9$) with a temperature of 0.7. This approach enables efficient construction of large-scale datasets, and is readily extensible to other existing corpora such as Li et al. (2024) and Cheng et al. (2024b). More details are presented in Appendix A.1.

### 3.2 BASELINES

We evaluate the following baselines: **(1) Plan-and-Solve (PS) Prompting:** It prompts the LLM to first devise a natural language plan to decompose the entire task into smaller subtasks, and then generate the response following the plan (Wang et al., 2023). **(2) Quiet-STaR:** It automatically learns implicit plans for generating each token from WebInstruct (Zelikman et al., 2024). **(3) Vanilla Training:** It first trains the LLM on WebInstruct and then prompts it to directly generate the response.

### 3.3 EXPERIMENTAL SETUP

We select Mistral-7B (Jiang et al., 2023) and Llama-2-7B/13B (Touvron et al., 2023) as our backbone models. For Mistral/Llama, we use a learning rate of 5e-6/1e-5 and a global batch size of 512/256, respectively. We set the maximum training epochs to 2. During inference, models are instructed to generate CoT-style responses for all tasks. Unless stated otherwise, we conduct evaluations under the

few-shot setting (from 2-shot to 4-shot) without fine-tuning on evaluation benchmarks. The above settings are also applied on baselines for fair comparisons.

## 3.4 EVALUATION BENCHMARKS

We evaluate CODEPLAN across a diverse range of tasks necessitating multi-step reasoning:

**Mathematical Reasoning.** It involves three benchmarks: (1) GSM8K (Cobbe et al., 2021), a collection of grade-school problems; (2) MATH (Hendrycks et al., 2021), a more challenging suite of high-school-level problems; and (3) SVAMP (Patel et al., 2021), a robustness evaluation benchmark.

**Symbolic Reasoning.** We use four benchmarks requiring multi-step logical deductions: (1) Boolean Expression from Big-bench-hard (BBH) (Suzgun et al., 2022), where the model infers the value of a complex boolean expression; (2) Coin Flipping (Wei et al., 2022b), which requires determining the final face after a sequence of flips. We use the challenging 4-flip version; (3) Last Letter Concatenation (Wei et al., 2022b), which requires concatenating the last letter of a 4-word sequence; and (4) Dyck Language from BBH, which requires predicting the closing parentheses of a Dyck-4 word. This task often demands over 10 reasoning steps. We find all models struggle to cahieve non-trivial few-shot performance on this task due to model degeneration (Holtzman et al., 2020). To mitigate these confounding factors beyond planning capabilities, we synthesize 5K examples based on official implementation to fine-tune each model before evaluation.

**Instruction-Following.** We assess the proficiency in following real-world instructions on two benchmarks: (1) AlpacaEval 1.0 (Li et al., 2023) and 2.0 (Dubois et al., 2024), which capture representative user interactions ; and (2) MT-Bench (Zheng et al., 2024), a meticulously curated set of high-quality queries spanning eight domains. Since our training data are not specifically tailored for instruction-following, we follow prior work (Tunstall et al., 2023) to continue fine-tuning each trained model on 150K randomly sampled examples from UltraChat (Ding et al., 2023).

**Multi-hop Question-Answering (QA).** We assess multi-step reasoning over complex information dependencies on two becnhmarks: (1) HotpotQA (Yang et al., 2018), which comprises 2-hop questions requiring reasoning over two supporting passages; and (2) MuSiQue (Trivedi et al., 2022), which consists of 2-hop to 4-hop questions with intricate dependency structures. Since our method focuses primarily on enhancing planning capabilities rather than knowledge acquisition, we provide the gold reference passages in the context during evaluation. We report the exact match (EM)scores.

**Decision-Making.** We use one benchmark to evaluate the performance in sequential decision-making scenarios: ALFWorld (Shridhar et al., 2020), a text-based virtual household environment comprising six distinct task types. It necessitates the model to navigate through intricate sequences of actions and observations, posing a rigorous test of the models' planning capabilities in dynamic environments. We use ReAct-style reasoning steps (Yao et al., 2022) during evaluation.

## 3.5 MAIN RESULTS

As highlighted in Table 3, **CODEPLAN shows consistent and substantial performance improvements across all backbone models and most benchmarks against the baselines**, underscoring the efficacy of incorporating code-form plans in enhancing LLMs' systematic reasoning capabilities. Specifically, CODEPLAN generally outperforms the PS Prompting baseline, often by a substantial margin. This indicates that the benefits of learning to plan in code are beyond what can be achieved by planning in natural language with careful prompt engineering alone. Quiet-STaR demonstrates inferior performance even compared to the backbone model despite significant computation overhead[1], illustrating the difficulty of learning implicit plans through latent variables. Notably, the vanilla training approach, which solely learns low-level reasoning steps, does not always improve backbone models' performance on downstream tasks. For instance, the performance of Mistral-7B drops by 2 to 9 points on several symbolic, multi-hop QA and decision-making tasks after vanilla training. We attribute this to the distribution gap between the massive, wide-ranging corpus and downstream benchmarks, thereby leading to suboptimal adaptation of the model's reasoning capabilities. **In contrast, CODEPLAN consistently improves upon the initial backbone model across all benchmarks, demonstrating its ability to develop more generalizable and transferable planning**

---

[1]We use the official implementation for Quiet-STaR, which only supports Mistral and not Llama series.

Table 3: Main results on five types of reasoning tasks. $\Delta$ means the margin between vanilla training (**Vanilla**) and CODEPLAN. On average, CODEPLAN yields a relative 25.1% performance gain against vanilla training. We highlight the best results in **bold**. We report accuracies for mathematical reasoning, symbolic reasoning, and decision-making tasks, the EM and F1 scores for multi-hop QA tasks, the win rate for AlpacaEval, and the GPT-4 score for MT-bench.

| Model | Mathematical Reasoning | | | Symbolic Reasoning | | | |
|---|---|---|---|---|---|---|---|
| | GSM8K | MATH | SVAMP | Boolean | Coin Flip | Last Letter | Dyck Language |
| **Mistral-7B** | 46.9 | 18.8 | 47.5 | 77.2 | 84.1 | 39.5 | 73.0 |
| **+ PS Prompting** | 45.5 | 17.3 | 58.5 | 75.6 | 79.5 | 41.5 | 70.1 |
| **+ Quiet-STaR** | 45.6 | 15.9 | 47.5 | 66.0 | 57.2 | 1.5 | 54.4 |
| **+ Vanilla** | 54.1 | 31.5 | 55.2 | 85.6 | 86.1 | 37.5 | 72.0 |
| **+ CODEPLAN** | **59.5** | **34.3** | **61.4** | **90.8** | **92.6** | **57.5** | **87.2** |
| $\Delta$ | +5.4 | +2.8 | +6.2 | +4.4 | +6.5 | +20.0 | +15.2 |
| (Relative Gain) | (+10.0%) | (+8.9%) | (+11.2%) | (+5.1%) | (+7.5%) | (+53.3%) | (+21.1%) |
| **Llama-2-7B** | 16.5 | 7.8 | 34.9 | 58.8 | 60.0 | 2.0 | 71.6 |
| **+ PS Prompting** | 12.0 | 4.7 | 27.6 | 61.2 | 61.4 | 1.0 | 70.4 |
| **+ Vanilla** | 30.7 | 19.6 | 36.6 | 75.2 | 54.4 | 0.0 | 63.6 |
| **+ CODEPLAN** | **33.8** | **20.8** | **41.5** | **79.2** | **63.0** | **5.0** | **88.0** |
| $\Delta$ | +3.1 | +1.2 | +4.9 | +4.0 | +8.6 | +5.0 | +16.4 |
| (Relative Gain) | (+10.1%) | (+6.1%) | (+13.4%) | (+5.3%) | (+15.8%) | (N/A) | (+25.8%) |
| **Llama-2-13B** | 30.2 | 9.9 | 41.9 | 72.4 | 85.5 | 1.0 | 65.2 |
| **+ PS Prompting** | 22.0 | 9.5 | 35.2 | 71.6 | 81.5 | **24.5** | 52.8 |
| **+ Vanilla** | 44.3 | 23.6 | 45.9 | 85.5 | 59.7 | 15.0 | 71.2 |
| **+ CODEPLAN** | **49.5** | **27.4** | **53.4** | **86.4** | **100.0** | 23.5 | **88.0** |
| $\Delta$ | +5.2 | +3.8 | +7.5 | +0.9 | +40.3 | +8.5 | +16.8 |
| (Relative Gain) | (+11.7%) | (+16.1%) | (+16.4%) | (+1.1%) | (+67.5%) | (+56.7%) | (+23.6%) |

| Model | Instruction-Following | | | Multi-hop QA | | Decision-Making |
|---|---|---|---|---|---|---|
| | AlpacaEval | | MT-Bench | MuSiQue | HotpotQA | ALFWorld |
| | 1.0 | 2.0 | | | | |
| **Mistral-7B** | 65.2 | 5.0 | 1.7 | 29.8 | 35.4 | 23.2 |
| **+ PS Prompting** | 56.7 | 4.9 | 6.0 | 36.2 | 35.8 | **25.2** |
| **+ Quiet-STaR** | 53.5 | 4.1 | 3.5 | 27.3 | 25.0 | 10.5 |
| **+ Vanilla** | 69.9 | 6.0 | 6.9 | 33.7 | 33.0 | 14.1 |
| **+ CODEPLAN** | **71.9** | **10.7** | **8.7** | **37.2** | **40.4** | 23.2 |
| $\Delta$ | +2.0 | +4.7 | +1.8 | +3.5 | +7.4 | +9.1 |
| (Relative Gain) | (+2.9%) | (+78.3%) | (+26.1%) | (+10.4%) | (+22.4%) | (+64.5%) |
| **Llama-2-7B** | 61.5 | 5.8 | 5.7 | 22.2 | 9.2 | 6.1 |
| **+ PS Prompting** | 46.0 | 4.4 | 4.9 | 11.5 | 10.6 | 10.1 |
| **+ Vanilla** | 58.0 | 3.8 | 5.8 | 25.0 | 16.0 | 12.1 |
| **+ CODEPLAN** | **65.1** | **6.2** | **6.2** | **27.4** | **27.4** | **14.1** |
| $\Delta$ | +7.1 | +2.4 | +0.4 | +2.4 | +11.4 | +2.0 |
| (Relative Gain) | (+12.2%) | (+63.2%) | (+6.9%) | (+9.6%) | (+71.3%) | (+16.5%) |
| **Llama-2-13B** | 66.7 | 6.5 | 6.1 | 26.8 | 38.0 | 23.2 |
| **+ PS Prompting** | 52.8 | 5.0 | 5.6 | 31.3 | 25.8 | 19.2 |
| **+ Vanilla** | 66.7 | 6.4 | 5.9 | 28.3 | 34.0 | 21.2 |
| **+ CODEPLAN** | **73.9** | **12.2** | **7.1** | **34.8** | **40.4** | **33.3** |
| $\Delta$ | +7.2 | +5.8 | +1.2 | +6.5 | +6.4 | +12.1 |
| (Relative Gain) | (+10.8%) | (+74.5%) | (+20.3%) | (+23.0%) | (+18.8%) | (+57.1%) |

**knowledge during training.** The superiority may result from the inherent structured, concise, and unambiguous semantics encoded within code-form plans. In this way, LLMs can more effectively extract and internalize the underlying planning signals in natural language data.

## 3.6 ANALYSIS

We analyze two key benefits of CODEPLAN: (1) increasing superiority on more complex problems (**Finding 1**) and (2) improved training data efficiency (**Finding 2**). Furthermore, we compare

CODEPLAN against two variants: planning in natural language (**Finding 3**) and reasoning through executable code (**Finding 4**), offering empirical evidence for the merits of leveraging code as intermediate plan representations. Appendix B also discusses the influence of plan annotation models, the efficiency of CODEPLAN, and case studies highlighting the strength and limitations of CODEPLAN.

**Finding 1: CODEPLAN Yields Amplified Benefits for Increasingly Complex Reasoning Tasks.** To gain deeper insights into the merits of CODEPLAN, we conduct experiments on MusiQue, which encompasses questions spanning various levels of complexity. As illustrated in Figure 2, CODEPLAN yields increasing performance gain as the task becomes more complex. The relative improvements in EM scores increase from 6.3% for 2-hop questions to a remarkable 43.8% for 4-hop questions.

This finding highlights a key insight – as reasoning challenges grow more intricate, the ability to generate and leverage structured code-form plans grows more valuable. For simple tasks,

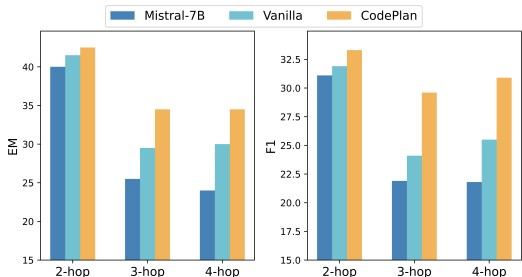

Figure 2: EM (**Left**) and F1 (**Right**) scores on the MuSiQue benchmark. $N$-hop means that the question requires $N$ reasoning steps to answer based on knowledge in Wikipedia passages.

LLMs' inherent language understanding capabilities often suffice. But as task complexity increases, the limitation of vanilla training—the ambiguity and obfuscation of planning signals in natural language data—becomes more clear. In such scenarios, CODEPLAN enables LLMs to systematically understand and frame the reasoning process, thereby navigating solution pathways more effectively.

**Finding 2: CODEPLAN Improves Data Efficiency.** We analyze the performance trajectories of CODEPLAN and vanilla training. As depicted in Figure 3, the LLM trained with CODEPLAN almost always outperforms its counterpart trained on the same raw prompt-response data, evincing superior knowledge acquisition and generalization to out-of-distribution reasoning challenges. Moreover, CODEPLAN achieves a more stable and consistent performance ascent, maintaining its supremacy throughout the training process. This showcases the advantage of CODEPLAN in developing transferable high-level reasoning skills from wide-ranging data, as illustrated by the case in Appendix B.4.

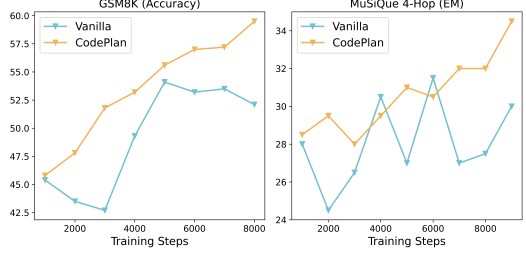

Figure 3: Performance trajectories on two downstream tasks via vanilla training and CODEPLAN. "4-Hop" denotes evaluating on the 4-hop subset.

**Finding 3: CODEPLAN Outperforms Planning in Natual Language.** To investigate how code-form plans compare to natural language plans, we conduct comparative experiments by replacing the code plans in our dataset with natural language counterparts[2]. Figure 4 shows the results. Training with natural language plans also leads to moderate improvement over the vanilla baseline. However, CODEPLAN consistently outperforms its natural language counterparts across all benchmarks by substantial margins. The performance gaps are particularly pronounced on complex reasoning tasks requiring structured planning, such as mathematical reasoning, symbolic reasoning, instruction-

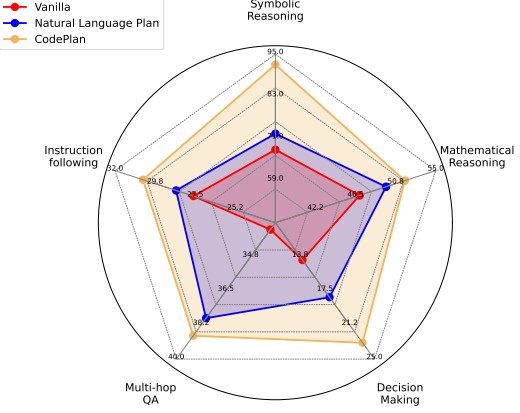

Figure 4: Comparing natural language planning with CODEPLAN. The scores of each type of task are averaged across all corresponding benchmarks.

---

[2]We generate natural language plans using the pipeline in §3.1 with a modified prompt in Appendix A.3.

Table 4: The negative log-likelihood of generating responses in different plan forms. Stage 1/2 refers to planning/surface realization, respectively. "Overall" is calculated by summing up the two parts.

| Method | Stage 1: $-\log p(Z^*|X)$ | Stage 2: $-\log p(Y|X, Z^*)$ | Overall |
|---|---|---|---|
| Vanilla | 0. | 0.689 | 0.689 |
| Natural Language Plan | 0.351 | **0.337** | 0.688 |
| CODEPLAN | **0.237** | 0.347 | **0.580** |

following, and decision-making tasks, with relative improvements of 4%, 27.2%, 6.5%, and 27.5%, respectively. This validates the superiority of code representations over natural language.

Additionally, we are curious about why planning in code outperforms natural language. We evaluate the performance of the two-stage process: planning and surface realization. For each plan form, Table 4 reports the model's negative log-likelihood (NLL) on a 10K subset of the training data. The vanilla baseline without explicit planning exhibits a high overall NLL, reflecting its difficulty in directly modeling the complex mapping from prompts to final responses. While natural language planning substantially reduces NLL for surface realization, it introduces a significant challenge in planning compared to CODEPLAN. We attribute this to two reasons: (1) code provides a more structured and precise representation of complex reasoning logic compared to natural language, thus offering more concise and easier-to-learn plan labels. (2) by framing plan generation as code generation, LLMs can better leverage their pre-training knowledge, as code is far more prevalent in pre-training corpora than natural language plans. Consequently, CODEPLAN achieves a more substantial overall NLL improvement, providing empirical validation for our theoretical framework in Eq. 4 that requires minimizing the combined difficulty of planning and surface realization.

**Finding 4: CODEPLAN Outperforms Executable Code-form Reasoning.** Prior work primarily focuses on using executable code to directly solve mathematical reasoning tasks (Gao et al., 2023). However, we posit that this approach is inherently limited when tackling broad, multi-step reasoning challenges that necessitate deep natural language understanding capabilities. To substantiate this claim, we conduct comparative experiments by replacing the code plans in our dataset with executable code solutions. Specifically, instead of constructing pseudocode plans, we directly translate each response into executable code that can output the answer after execution with a code interpreter, using the prompt in Appendix A.3. We refer to this baseline as CODEREASON.

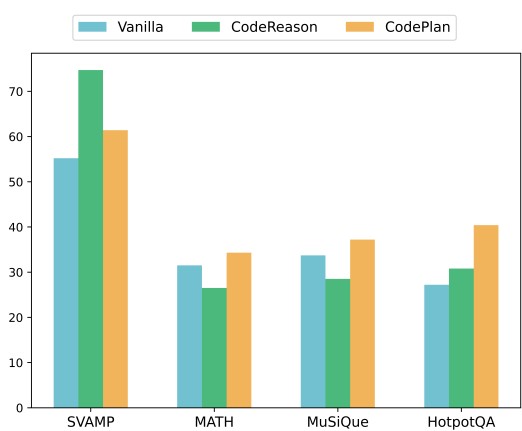

Figure 5: Comparing CODEREASON (i.e., executable code-form reasoning) with CODEPLAN.

As presented in Figure 5, while CODEREASON achieves the highest performance on SVAMP, a benchmark comprised of relatively simple math word problems, it consistently underperforms CODEPLAN across a diverse array of reasoning benchmarks that demand extensive natural language understanding capabilities, such as MATH, MuSiQue, and HotpotQA. Notably, CODEREASON even lags behind the vanilla training baseline on MATH and MuSiQue. In contrast, CODEPLAN's novel framework of generating code plans as an intermediate representation seamlessly integrates robust planning capabilities with the rich language understanding abilities of LLMs, yielding superior performance on intricate multi-step reasoning challenges.

## 4 RELATED WORK

**LLMs for Reasoning.** Endowing LLMs with robust reasoning abilities remains a formidable challenge. Existing approaches predominantly fall into three categories: (1) **Prompting techniques**, which use expert-designed prompts to elicit reasoning skills without training (Wei et al., 2022b; Press et al., 2023; Imani et al., 2023; Hong et al., 2024) (2) **Task-specific fine-tuning**, which curates tailored fine-tuning data or rewards to improve reasoning in specific tasks like mathematical reasoning (Yu et al., 2023; Mitra et al., 2024; Shao et al., 2024; OpenAI, 2024), code reasoning (Le et al., 2022; Shen et al., 2023), instruction-following (Cui et al., 2023), visual reasoning (Cheng et al., 2024a), and decision-making (Zeng et al., 2023; Guan et al., 2024) tasks. However, these approaches often falter in generalizing beyond their intended tasks. (3) **Tool integration**, which seeks to augment LLMs' reasoning capabilities by incorporating external tools like calculators (Schick et al., 2024), retrievers (Asai et al., 2024), and code interpreters (Gao et al., 2023). Compared to these works, CODEPLAN focuses on enhancing LLMs' high-level planning capabilities in diverse domains, without relying on task-specific prompts, fine-tuning data, rewards, or tools.

**Deliberate Planning in LLMs.** Enhancing LLMs' planning capabilities is crucial for complex reasoning tasks (Yang et al., 2023). Prior work primarily focuses on teaching LLMs to plan in natural language via task-specific prompts (Wang et al., 2023; Khot et al., 2022) or curated fine-tuning data (Yin et al., 2024; Guan et al., 2024). In contrast, CODEPLAN innovatively introduces code as a structured and versatile plan representation. Recent works also attempt to learn implicit planning (e.g., latent code or verbal words) from wide-ranging text corpora (Zelikman et al., 2024; Cornille et al., 2024). However, these approaches may struggle to automatically unveil effective planning signals from the vast space and often introduce significant computation overhead during training due to online sampling over prior and posterior distributions. In contrast, CODEPLAN introduces neglectable computation cost as illustrated in Appendix B.3. Additionally, recent works also explore multi-path planning (Yao et al., 2024) and iterative plan refinement (Shinn et al., 2024), which are orthogonal to our work. We leave integrating CODEPLAN with such techniques for future work.

**Code-aided Reasoning.** Recent works have explored leveraging code to empower LLMs for complex reasoning. One approach directly employs prompting techniques (Gao et al., 2023; Ye et al., 2023) or curated fine-tuning data (Gou et al., 2023; Zhou et al., 2023) to generate executable code as a surrogate for natural language response, subsequently utilizing a code interpreter to derive the answer. Despite the precision afforded by executing code, this framework suffers from limited data scalability and is significantly limited to narrow domains such as mathematical calculation. Beyond direct problem-solving, code has also been utilized to enhance LLMs' capabilities in handling structured reasoning tasks, such as graph generation (Madaan et al., 2022), event structure prediction (Wang et al., 2022), and decision-making (Wang et al., 2024) tasks. Sharing a similar motivation, our work leverages code to represent intricate reasoning structures.

## 5 CONCLUSION

In this work, we introduce a pioneering framework to endow LLMs with robust planning capabilities through the explicit supervision of code-form plans. By reframing plan generation as a code generation task, CODEPLAN harnesses the structured and versatile nature of code to capture the rich semantics and control flows underpinning sophisticated reasoning processes. Importantly, CODEPLAN allows the automatic extraction of code-form plans from massive, wide-ranging text corpora without the need for curated, task-specific datasets. We demonstrate the effectiveness of this framework by training CODEPLAN on a large-scale dataset comprised of 2M natural language problems paired with their corresponding code-form plans. Across an extensive evaluation spanning 13 challenging multi-step reasoning benchmarks, CODEPLAN demonstrates remarkable efficacy, consistently and substantially outperforming vanilla training by a substantial margin. In-depth analyses further corroborate CODEPLAN's increasing performance gain on complex problems and generalization ability. This work paves the way for several promising research directions, including exploring diverse posterior distributions over plans, enabling multi-path planning, facilitating plan verification, reflection and refinement, and realizing agents that can seamlessly leverage external knowledge sources and APIs within their planning and reasoning processes.

## 6 ACKNOWLEDGEMENTS

This work was supported by the National Science Foundation for Distinguished Young Scholars (with No. 62125604), and he Tsinghua University Initiative Scientific Research Program. This work was also supported by the Ant Group Research Intern Program.

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

# A  IMPLEMENTATION DETAILS

## A.1  TRAINING DATA CURATION DETAILS

The curation of high-quality training data is crucial for the success of our approach. To ensure the integrity and relevance of the generated code-form plans, we filter examples without valid terminations in the plans, as this is essential for maintaining logical coherence. Figure 6 illustrates the pipeline for curating the training data.

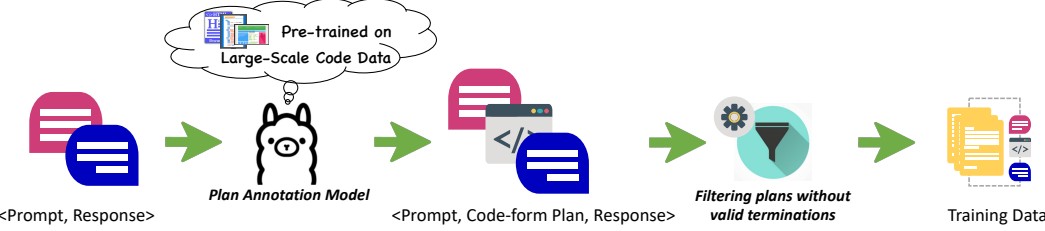

Figure 6: Training data curation pipeline.

Furthermore, we explored several fine-grained filtering mechanisms to refine our dataset. One notable approach involved assessing the information gain of each plan $Z^*$ by comparing the perplexity of generating $Y$ with and without the plan (i.e., $P(Y|X)$ vs. $P(Y|X, Z^*)$). However, our pilot experiments revealed that these additional filtering steps yielded only marginal improvements in the performance of LLMs. In the interest of efficiency and to avoid potential overfitting, we opted to omit these more complex filtering mechanisms in our final process. Appendix B.2 shows more discussion about plan quality.

## A.2  DATA STATISTICS

Table 5 and Table 6 present the detailed statistics of our training data and evaluation benchmarks, respectively. For mathematical reasoning tasks, we use the full test set of GSM8K and SVAMP, and a non-calculation-intensive subset of MATH, including "Prealgebra," "Geometry," "Counting & Probability" since we focus on planning. For symbolic reasoning tasks, we use the official test set of Boolean and Dyck Language, and use the official code to generate a challenging 4-hop test set for Coin Flipping and Last Letter Concatenation. For instruction-following tasks, we use the official test set of AlpacaEval and MT-Bench. For multi-hop QA tasks, we randomly sample 200 questions from the 2/3/4-hop subset of MuSiQue (600 in total) and randomly sample 500 questions from HotpotQA. For the decision-making task, we use four task types, including "pick&place," "clean&place," "heat&place," and "cool& place."

Table 5: Statistics of the curated training data.

| Aspects | # Examples | Avg. Prompt Length | Avg. Plan Length | Avg. Response Length |
|---|---|---|---|---|
| **Statistics** | 2,335,072 | 36.2 | 45.7 | 134.7 |

Table 6: Number of test examples in the evaluation benchmarks.

| Mathematical Reasoning GSM8K/MATH/SVAMP | Symbolic Reasoning Boolean/Coin Flip/Last Letter/Dyck Language | Instruction-Following AlpacaEval 1.0/2.0/MT-Bench | Multi-hop QA MuSiQue/HotpotQA | Decision-Making ALFWorld |
|---|---|---|---|---|
| 1,319/1,824/1,000 | 250/1,000/1,000/250 | 805/805/80 | 600/500 | 100 |

## A.3  PROMPTS

**Prompt for Annotating Natural Language Plans.**  We present the prompt for annotating the natural language plans for the original WebInstruct data of prompt-response pairs in Table 7.

Table 7: Instruction for generating the natural language plan for a given prompt-response pair.

```
Prompt: {{Prompt}}
Response: {{Response}}

Given a prompt-response pair, your task is to describe the high-level logic of the response
using natural language.  Such that following this logic, models can easily generate the
response.

The logic should balance conciseness and informativeness.
The logic should be high-level, instead of replicating low-level details in the response.
The logic should be less than 200 words (adjust its length based on response lengths).
```

**Prompt for Annotating Executable Code for Reasoning.** We present the prompt for annotating executable codes for the original training data in Table 8.

Table 8: Instruction for generating the executable code for a given prompt-response pair.

```
Prompt: {{Prompt}}
Response: {{Response}}

Given a prompt-response pair, your task is to convert a natural language response into an
executable Python code that can print the same response.

The execution output should be consistent with the natural language answer.
```

# B  ADDITIONAL RESULTS

## B.1  EXPERIMENTS ON LLAMA-3 MODELS

## B.2  ASSESSING THE ROBUSTNESS OF CODEPLAN TO PLAN QUALITY DRIFT

To ensure the reliability and generalizability of our approach, it is crucial to assess the robustness of CODEPLAN with respect to various factors that could potentially influence its performance. Specifically, we focus on the choice of plan annotation models. This analysis aims to provide insights into the stability of our method across different experimental conditions.

We initially used Llama-3-8B-Instruct to construct code-form plans from natural language responses, balancing efficiency and performance. To investigate the impact of label quality on CODEPLAN, we conducted an ablation study using two additional models for plan annotation: Gemma-2B-instruct[3] and Llama-3-70B-instruct (Dubey et al., 2024). These models represent a spectrum of capabilities, allowing us to assess how variations in label quality affect overall performance.

Table 9: Performance of CODEPLAN on three representative types of reasoning tasks when using different models for annotating code-form plans. Scores are averaged across corresponding benchmarks for each task.

| Annotation Model | Gemma-2B-Instruct | Llama-3-8B-Instruct | Llama-3-70B-Instruct |
|---|---|---|---|
| **Mathematical Reasoning** | 45.1 | 51.7 | 50.8 |
| **Multi-hop QA** | 29.9 | 38.8 | 37.7 |

Table 9 presents the results of this comparative analysis. While there is a substantial performance gap between Gemma-2B-Instruct and Llama-3-8B-Instruct, the difference between Llama-3-8B-Instruct and the more powerful Llama-3-70B-Instruct is relatively small across all reasoning tasks. This suggests that once the annotation model reaches a sufficient level of competence, CODEPLAN's performance becomes largely insensitive to further improvements in plan quality, underscoring its stability and resilience. The robustness of CODEPLAN to variations in plan quality offers several key advantages. First, it alleviates the need for employing the most computationally expensive and resource-intensive models for plan annotation, enabling more efficient data curation and training.

---

[3]https://huggingface.co/google/gemma-2b-it

Table 10: Efficiency of CODEPLAN compared with vanilla training.

| Model | Memory (GB) | Time (Second) |
|---|---|---|
| **Vanilla** | 30G | 0.055 |
| **CODEPLAN** | 30G | 0.058 |

Table 11: One example for the mathematical reasoning task (from the MATH benchmark).

Prompt:

There is a number $C$. The number $C$ is a multiple of 7, but is NOT a multiple of 3. The number $C$ is greater than 20, but less than 30. What is this number $C$?

Plan and Response from CODEPLAN:

```python
def find_number_c():
    multiples_of_7 = [i for i in range(21, 30) if i % 7 == 0]
    multiples_of_3 = [i for i in range(21, 30) if i % 3 == 0]
    result = [i for i in multiples_of_7 if i not in multiples_of_3]
    return result[0]
```

The multiples of 7 between 20 and 30 are 21 and 28. The multiples of 3 between 20 and 30 are 21, 24, 27, and 30. Therefore, the number $C$ is 28. The answer is 28.

Response from Vanilla:

The number $C$ is a multiple of 7, but is NOT a multiple of 3. This means that $C$ is a multiple of 21, but is NOT a multiple of 27. The only number that fits this description is 21. The answer is 21

Second, it enhances the generalizability of our approach, as CODEPLAN can maintain its effectiveness even when the plan annotations deviate from perfection due to factors such as domain shift or annotation noise.

### B.3 EFFICIENCY ANALYSIS

For efficiency evaluation, Table 10 reports the memory usage, and average per-example inference time of CODEPLAN and the vanilla training baseline. During evaluating, we set the batch size to 1 and use one A100 GPU. We calculate memory consumption using PyTorch toolkits pytorch (2023). The per-example inference time is averaged over 1000 generations.

The results demonstrate the comparable computational efficiency of CODEPLAN with vanilla training, with only a negligible increase in inference memory usage. Moreover, the average inference time of CODEPLAN is only marginally higher ($\sim$5%) than that of vanilla training. These efficiency metrics highlight that CODEPLAN does not sacrifice much computation overhead to get substantial performance improvements, since we use pseudocode to concisely outline the reasoning structures.

### B.4 CASE STUDY

To provide deeper insights into how CODEPLAN enhances LLM reasoning capabilities, we present a detailed case study. This analysis involves the testing benchmarks used in the main paper, the reusability of functions in the code-form plans to explain the high data efficiency of CODEPLAN, as well as the strong performance of CODEPLAN for two popular tests, including *Value Comparison* and *Letter Counting*, thus comprehensively illustrating the concrete benefits of our code-based planning approach across diverse reasoning challenges.

**Cases for Benchmarks.** We present one example for each type of task in Table 11, 12, 13, 14, respectively. Note that the example for the instruction-following task has been shown in Figure 1 in the main paper. As exemplified, the code-form plan clearly outlines the step-by-step reasoning process:

- **Mathematical Reasoning** (Table 11): This example requires identifying a number that satisfies multiple constraints. The vanilla pre-trained model fails to correctly incorporate all conditions, yielding an incorrect answer. In contrast, CODEPLAN's code plan methodically enumerates the relevant number ranges, applies the given criteria through logical operations,

Table 12: One example for the symbolic reasoning task (from the Last Letter Concatenation benchmark).

---

`Prompt:`
Take the last letters of the words in "Ofe Aliza Betzy Rohan" and concatenate them.

`Plan and Response from CODEPLAN:`

```python
def extract_and_concatenate_last_letters(input_str):
    words = input_str.split()
    concatenated_result = ""
    for word in words:
        last_letter = word[-1]
        concatenated_result += last_letter
    return concatenated_result
input_str = "Ofe Aliza Betzy Rohan"
result = extract_and_concatenate_last_letters(input_str)
```

Current word is "Ofe", last letter is "e", concatenated result is "e". Current word is "Aliza", last letter is "a", concatenated result is "ea". Current word is "Betzy", last letter is "y", concatenated result is "eay". Current word is "Rohan", last letter is "n", concatenated result is "eayn". The answer is eayn.

`Response from Vanilla:`
Current word is "Ofe", last letter is "e", concatenated result is "e". Current word is "Aliza", last letter is "a", concatenated result is "ea". Current word is "Betzy", last letter is "y", concatenated result is "ey". Current word is "Rohan", last letter is "n", concatenated result is "eya". The answer is eaeya.

---

Table 13: One example for the multi-hop QA task (from the MuSiQue benchmark).

---

`Prompt:`
The beautiful country used as a backdrop for the movie The Lord of the Rings was founded on what date?

`Plan and Response from CODEPLAN:`

```python
def get_answer():
    movie = query("What movie is used as a backdrop") = "The Lord of the Rings"
    country = query("What country is used as a backdrop") = "New Zealand"
    event = query("What event is related to the country") = "Treaty of Waitangi"
    date = query("What is the date of the event") = "6 February 1840"
    answer = date
    return answer
```

The movie The Lord of the Rings was filmed in New Zealand. The Treaty of Waitangi was signed in New Zealand on 6 February 1840. The answer is 6 February 1840.

`Response from Vanilla:`
he movie The Lord of the Rings was released in 2001. The country was founded in 1840. The answer is 1840.

---

and precisely identifies the correct solution. The code naturally captures the step-by-step reasoning process, breaking down the complex problem into interpretable sub-tasks.

- **Symbolic Reasoning** (Table 12): Here, the task involves concatenating the last letters of words in a given string. While the vanilla model makes a mistake in tracking the concatenation order, CODEPLAN's code plan clearly delineates the iterative process of extracting each word's last letter and appending it to the result string. The structured nature of the code ensures precise execution of the required operations, leading to the correct solution.

- **Multi-hop QA** (Table 13): Answering this question requires reasoning over multiple pieces of information and making implicit connections. The vanilla model struggles to synthesize the relevant facts, providing an incorrect response. However, CODEPLAN's code plan explicitly outlines the necessary steps: identifying the key entities mentioned, querying relevant information about each, and logically combining the retrieved knowledge to derive the final answer. This structured approach enables CODEPLAN to navigate the multi-hop reasoning process successfully.

- **Decision-Making** (Table 14): In this task set in a simulated environment, the agent must execute a sequence of actions to accomplish the given goal. While the vanilla-trained LM struggles to maintain a coherent plan of action, CODEPLAN's code-form plan clearly specifies the high-level steps required, such as locating and obtaining the target object, cooling it, and placing it in the designated location. This explicit planning stage allows CODEPLAN to systematically break down the complex task and execute the necessary actions to achieve the desired outcome.

Table 14: One example for the decision-making task (from the ALFWorld benchmark).

```
Prompt:
```
Cool a mug and put it in coffeemachine

```
Plan and Response from CODEPLAN:
```

```
def cool_mug_put_in_coffeemachine():
    for receptable in recept_to_check:
        observation = goto(receptable)
        if 'closed' in observation:
            observation = open(receptable)
        if "mug" in observation:
            found_mug = take("mug", receptable)
    goto("fridge")
    cool(found_mug, "fridge")
    goto("coffeemachine")
    put(found_mug, "coffeemachine")
```

Act: think: To solve the task, I need to find and take a mug, then cool it with fridge, then put it in coffeemachine.
Obs: OK.
Act: think: First I need to find a mug. A mug is more likely to appear in drawer (1-5), cabinet (1-16), countertop (1-3), sinkbasin (1), stoveburner (1-4), fridge (1), toaster (1), shelf (1-3). I can check one by one, starting with drawer 1.
Obs: OK.
Act: go to drawer 1
Obs: The drawer 1 is closed.
....
Act: go to countertop 1
Obs: On the countertop 1, you see a apple 1, a creditcard 2, a mug 2, and a spatula 1.
Act: think: Now I find a mug 2. Next, I need to take it.
Obs: OK.
Act: take mug 2 from countertop 1.
Obs: You pick up the mug 2 from countertop 1
Act: think: Now I take a mug 2. Next, I need to go to a fridge 1 and cool it.
Obs: OK.
Act: go to fridge 1
Obs: The fridge 1 is closed.
Act: cool mug 2 with fridge 1
Obs: You cool the mug 2 using the fridge 1.
Act: think: Now I cool a mug (1). Next, I need to put it in/on coffeemachine 1.
Obs: OK.
Act: go to coffeemachine 1
Obs: On the coffeemachine 1, you see nothing.
Act: put mug 2 in/on coffeemachine 1
Obs: You put the mug 2 in/on the coffeemachine 1.

```
Response from Vanilla:
```
Act: think: To solve the task, I need to find and take a cool mug, then put it in coffeemachine.
Obs: OK.
Act: think: First I need to find a cool mug. A cool mug is more likely to appear in fridge (1), countertop (1-3), cabinet (1-4), drawer (1-3). I can check one by one, starting with fridge 1.
Obs: OK.
...
Act: take mug 2 from countertop 1 Obs: You pick up the mug 2 from the countertop 1.
Act: think: Now I find a cool mug 2. Next, I need to go to a coffeemachine 1
Obs: OK.
Act: go to coffeemachine 1
Obs: Nothing happens.
Act: put mug 2 in coffeemachine 1
Obs: Nothing happens.
Act: think: Now I put a cool mug (2). Next, I need to turn on coffeemachine 1.
Obs: OK.
Act: turn on coffeemachine 1
Obs: Nothing happens.
Act: think: Now I turn on a coffeemachine (1). Next, I need to wait until it is done.
Obs: OK.
...
Act: think: Now I wait until a coffeemachine (1) is done. Next, I need to take a mug (2).
Obs: OK.
...
Act: think: Now I put a mug (2) in/on a dishwasher (1). Next, I need to turn on a dishwasher (1). Obs: OK.
...

Through these diverse examples, we observe that CODEPLAN's code-based plans consistently provide clear, structured representations of the reasoning process across various task types. The formal semantics of code enable the explicit encoding of intricate logic, control flow, and hierarchical task decompositions. This explicit planning stage allows the model to methodically navigate complex problems, leading to more accurate, reliable, and interpretable solutions compared to direct response generation without structured planning.

Table 15: Two examples from the training data of CODEPLAN, where two different questions can be solved through similar code-form plans.

| Prompt 1: | Prompt 2: |
|---|---|
| Which numbers in the following set are prime numbers: (4,7,31, 117, 324)? | What is the 10th prime number greater than 50, if they are listed in order? |
| Plan and Response for Prompt 1: | Plan and Response for Prompt 2: |

```python
def is_prime(n):
    if n <= 1:
        return False
    for i in range(2, int(n ** 0.5) + 1):
        if n % i == 0:
            return False
    return True
def find_prime_numbers(numbers):
    prime_numbers = []
    for num in numbers:
        if is_prime(num):
            prime_numbers.append(num)
    return prime_numbers
numbers = [4, 7, 31, 117, 324]
prime_numbers = find_prime_numbers(numbers)
print(prime_numbers)
```

A prime number is a natural number greater than 1 that is only divisible by 1 and itself.
From the given set, 4 and 324 are not prime numbers because they are even.
117 is not a prime number because it can be written as $117 = 3x39 = 3x3x13 = 3^2x13$, which means it has factors other than 1 and itself.
Therefore, the only prime numbers in the set are 7 and 31.

```python
def find_10th_prime_greater_than_50():
    primes = []
    num = 53
    while len(primes) < 10:
        if is_prime(num):
            primes.append(num)
        num += 1
    return primes[-1]
def is_prime(n):
    if n <= 1:
        return False
    for i in range(2, int(n ** 0.5) + 1):
        if n % i == 0:
            return False
    return True
```

The prime numbers greater than 50 are: 53, 59, 61, 67, 71, 73, 79, 83, 89, 97, ...The 10th prime number greater than 50 is 97.

**Cases for Reusable Functions.** One of the key advantages of CODEPLAN 's code-form planning is the ability to reuse functions across different instances. By abstracting certain common operations into reusable code blocks, CODEPLAN can achieve higher data efficiency. For instance, when tackling tasks involving mathematical reasoning, frequently used operations can be encapsulated in distinct functions that are reusable across multiple examples, as illustrated in Table 15.

**Cases for "Value Comparison" and "Letter Counting."** Recent studies showed that value comparison and letter counting are difficult even for GPT-4o. The former requires determining the larger one of two digits (e.g., comparing 9.8 and 9.11), while the latter requires counting the number of occurrences of a specific letter within a word, such as counting the number of "r"s in "strawberry." LLMs often falter when asked to handle these seemingly simple tasks because they lack a structured process for dealing with multi-part numerical representations or iterating through the word and applying conditions.

In contrast, CODEPLAN 's code-form plan effectively disambiguates the tasks by explicitly defining how to compare values and count letters, as shown in Table 16. For example, when comparing 9.8 and 9.11, CODEPLAN generates a code-form plan that systematically parses and compares each part of the numbers, ensuring that the comparison logic is clear and correct. If asked to count the number of "r"s in "strawberry," CODEPLAN produces a pseudocode plan that explicitly tracks each letter and increments the count when the specified letter is found.

## B.5 ERROR ANALYSIS

While CODEPLAN demonstrates significant improvements in reasoning capabilities, it is crucial to conduct a thorough error analysis to identify areas for future research and refinement. We categorize and examine three primary types of errors observed in the code plan generation and execution process:

- **Subtask Execution Error** (Table 17): This error occurs when the model correctly generates an appropriate high-level plan but fails to execute individual subtasks accurately. For example, in a multi-step math problem, the plan might correctly outline the sequence of operations, but an arithmetic error in one step leads to an incorrect final answer. Table 17 presents another error case from the Last Letter Concatenation benchmark. Here, the plan is correct, but an error in the reasoning path leads to an incorrect final result. This type of

Table 16: Examples from OpenAI's o1-preview with and without code-form planning for the "Value Comparison" task and "Letter Counting" task.

| Prompt for ``Value Comparison'': 9.8 and 9.11 which is larger? | Prompt for ``Letter Counting'': how many r is there in "strawberry" |
| --- | --- |
| Response with Code-form Planning: | Response with Code-form Planning: |

```
def compare():
    delta = 9.8 - 9.11
    if delta > 0.
        answer = '9.8 is larger'
    else:
        answer = '9.11 is larger
```

```
def get_frequenc( ) :
    cnt =0
    for c in "strawberry":
        if c="r":
            cnt += 1
    return cnt
```

delta = 9.8 - 9.11= 0.69. Since 0.69 > 0, 9.8 is larger.

There are 3 "r"s in the word "strawberry"

| Response without Planning: 9.11 is larger. | Response without Planning: The word "strawberry" contains two letter "r"s. Here's the breakdown: S T R A W B E R R Y The two "r"s are located in the third and eighth positions. |

Table 17: One example from the Last Letter Concatenation benchmark, which exhibits the subtask execution error highlighted in red.

Prompt:
Take the last letters of the words in "Nguyec5n Danish Aminta Javon" and concatenate them.

Plan and Response from CODEPLAN:

```
def extract_and_concatenate_last_letters(input_str):
    words = input_str.split()
    concatenated_result = ""
    for word in words:
        last_letter = word[-1]
        concatenated_result += last_letter
    return concatenated_result
input_str = "Ofe Aliza Betzy Rohan"
result = extract_and_concatenate_last_letters(input_str)
```

Current word is "Nguyec5n", last letter is "n", concatenated result is "n". Current word is "Danish", last letter is "i", concatenated result is "ni". Current word is "Aminta", last letter is "a", concatenated result is "nia". Current word is "Javon", last letter is "n", concatenated result is "nian". The answer is nian.

Table 18: One example from the AlpacaEval benchmark, which exhibits the plan following error highlighted in red.

Prompt:
Please read the paragraph below and count how many times the words "Amazon", "river", and "you" appear.
Please present the results in the format of "word, number of appearances" with each word on a separate line. Sort the lines in order of the number of appearances.

The Amazon, a mesmerizing expanse of nature's wonders, is home to the legendary Amazon River. Flowing through awe-inspiring landscapes like the Amazon rainforest, the river weaves its way through Brazil, Colombia, and Peru, giving life to countless creatures. From the mighty jaguars prowling the Amazon jungle to the vibrant macaws soaring above the canopy, this remarkable region teems with biodiversity...

Plan and Response from CODEPLAN:

```
def count_words(text):
    words = ["Amazon", "river", "you"]
    results = {}
    for word in words:
        results[word] = 0
    for sentence in text.split("."):
        for word in words:
            if word in sentence:
                results[word] += 1
    sorted_words = sorted(results.items(), key=lambda x: x[1], reverse=True)
    return "\n".join([f"{word}: {count}" for word, count in sorted_words])
```

Amazon, 20
river, 13
you, 1

error suggests that while CODEPLAN enhances high-level reasoning, there is still room for improvement in the performance of low-level reasoning steps.

Table 19: One Example from the MATH benchmark, which exhibits the plan error.

Prompt:

February 1, 2008 is a Friday. The last Friday of February 2008 will be February $x$, 2008. If 2008 is a leap year, what is the value of $x$?

Plan and Response from CODEPLAN:

```python
def find_last_friday_of_february(year):
    if is_leap_year(year):
        days_in_february = 29
    else:
        days_in_february = 28
    last_friday = days_in_february - (days_in_february % 7)
    return last_friday
def is_leap_year(year):
    return year % 4 == 0 and (year % 100 != 0 or year % 400 == 0)
year = 2008
x = find_last_friday_of_february(year)
print(x)
```

Since 2008 is a leap year, February has 29 days. Since 1 is a Friday, $29 - 29\%7 = 28$ is a Friday. Thus, $x = 28$. The answer is 28

- **Plan Following Error** (Table 18): This error type reveals potential disconnects between the planning and realization stages. In Table 18, the model generates a correct plan but deviates from it during the execution phase, omitting crucial steps. Addressing this error type could involve strengthening the coupling between planning and execution phases during training.

- **Plan Error** (Table 19): The most fundamental type of error occurs when the generated plan itself is flawed or incomplete. This indicates limitations in the model's ability to formulate comprehensive strategies for complex problems. Consider the example in Table 19 from a mathematical reasoning task. The generated plan incorrectly calculates "days in february % 7", which should be "(days in february -1 ) % 7". This type of error suggests that further refinement of the planning mechanism is necessary, particularly for tasks requiring nuanced multi-step reasoning.

Our analysis reveals that while CODEPLAN significantly enhances reasoning capabilities, there remain opportunities for improvement across various aspects of the planning and execution process. The subtask execution errors highlight the need for enhanced numerical precision and robustness in low-level computations. Plan following errors suggest potential benefits from stronger integration between the planning and realization stages during training. Finally, plan errors underscore the importance of further refining the model's ability to generate comprehensive and nuanced strategies for complex reasoning tasks.

