# OpenReview forum: "CodePlan: Unlocking Reasoning Potential in Large Language Models by Scaling Code-form Planning"
_ICLR.cc/2025/Conference — ICLR 2025 Poster_

### Official Review · Reviewer_k1j8 · 2024-11-01

**Soundness:** 2
**Presentation:** 3
**Contribution:** 2
**Rating:** 3
**Confidence:** 4

**Summary:**

This paper introduces CODEPLAN, which formulates planning for complex problems in the form of code. To enable the model to acquire such ability, the authors have existing LLMs to generate this code-style planning for existing samples as training data. The authors have conducted experiments on various reasoning tasks and have achieved improvements.

**Strengths:**

- The paper demonstrates performance improvements over baselines across different datasets.
- The authors have validated the effectiveness of using code as a form of planning.
- The authors have validated the effectiveness of using existing LLMs to generate code-style planning.

**Weaknesses:**

1. The font size in Table 1 is too small.
2. Overall, I believe this paper would be more suitable as a technical report or a short article. The idea of using code form for planning is a useful trick, but it lacks enough innovation:
 - It only addresses the issue of the **form** of planning, without fundamentally enhancing the model's planning and reasoning capabilities
 - A widely held assumption in the community is that code is important for LLMs to learn complex reasoning and long-text abilities. There is already an amount of work on this topic. Therefore, the techniques and conclusions of this paper are not surprising.
 - Under code-style planning, the core issue is how to generate effective code-style planning for a query from a specific domain or a general domain. However, this paper does not actually address this core issue. The method of generating through prompting shown in Table 2 is overly simplistic.
3. This method is constrained by the effectiveness of the LLM which generates code plannings for training. A common approach is to improve the quality of the training data through code execution results or log information. I suggest the author to consider this optimization method.
4. Since the paper posits that code-form plans are superior to other forms (such as text, first-order logic, graph, etc.), the authors should provide direct comparative experiments.

**Questions:**

Please refer to weaknesses.

---

> ### Author Response · Authors · 2024-11-20
> **Response to Reviewer k1j8 (1/2)**
>
> Thanks for the valuable comments.
>
> >  Weakness 4: Regarding the comparison to other plan forms such as text
>
> We have indeed provided comparative experiments between code-form plans and natural language plans, as shown in Figure 4 and Table 4 in Finding 3 of Section 3.6. These results demonstrate the superior performance of code-form plans across various reasoning tasks. Please see General Response 2 for a more detailed discussion.
>
> > Weakness 2(2): Regarding the effectiveness of code for complex reasoning is a widely held assumption
>
> While code is widely **hypothesized** to improve reasoning abilities, prior work mainly validates this hypothesis on math word problems ([1][2]) or code generation tasks ([3]), some work also finds code training may reduce performance in domains beyond code due to distribution shits ([4]). In contrast, to the best of our knowledge, our paper provides the **first systematic study** showing that introducing explicit **code-form planning** can lead to **consistent improvements across 13 reasoning tasks**, spanning mathematical reasoning, symbolic reasoning, instructionfollowing, multi-hop QA, and decision-making tasks. Furthermore, CodePlan can leverage the automatically constructed data from massive, wide-ranging text corpora.  The scalability is also a key differentiator from previous approaches, which usually rely on curated, task-specific datasets ([2]).
>
> >  Weakness 2 (1): Regarding the novelty of code-form planning
>
> We respectfully disagree that code-form planning lacks sufficient innovation. The development of novel data representations to enhance model knowledge adaption or learning is a significant area of research, as exemplified by several impactful studies:
> - Chain-of-Thought (CoT) prompting [5] introduced step-by-step reasoning “form”
> - Recent works [1][6] demonstrated the effectiveness of code “form” in capturing mathematical reasoning logic and graph reasoning structures, respectively.
> These approaches have garnered substantial attention and have influenced subsequent research. CodePlan extends these ideas by leveraging code-form plans to capture complex, multi-step reasoning processes across a diverse range of tasks. We believe these contributions represent a meaningful step forward in enhancing language models' reasoning abilities.
>
> “What fundamentally enhances model’s planning and reasoning capabilities” or “Whether LMs can plan and reason” are often meta-philosophical debates. Technically, we believe supervising LMs with **explicit, structured (code-form), scalable planning signals**, instead of merely low-level reasoning steps, is one key step towards this long-term goal.
>
> > Weakness 1 (3): Regarding optimizing code-style planning
>
> We acknowledge that optimizing code-style plan generation for specific domains is indeed an important area of research. However, this paper focuses on addressing a more fundamental question: can we scalably construct code-form plans from massive text corpora and improve LM performance across a wide range of reasoning tasks? By establishing this foundational understanding, our work paves the way for future advancements in this field. This is similar to how the initial demonstration of CoT prompting [5] laid the groundwork for subsequent investigations into advanced CoT realizations [7][8].
>
> > Weakness 3: Regarding optimizing code-style planning through code execution
>
> Thanks for the suggestion! However, measuring the plan quality of a training example (i.e., its influence on the final model performance) is still an open problem ([9][10]). For example, as indicated by our results in Table 4, more verbose plans do not lead to higher performance since they also pose a greater challenge for planning generation. Similarly, an executable code plan does not necessarily lead to higher performance than its non-executable counterpart.
>
> [1] PAL: Program-aided Language Models. ICML 2023
>
> [2] Tora: A tool-integrated reasoning agent for mathematical problem solving. ICLR 2023
>
> [3] At Which Training Stage Does Code Data Help LLMs Reasoning? ICLR 2024
>
> [4] DeepSeekMath: Pushing the Limits of Mathematical Reasoning in Open Language Models. arXiv 2024
>
> [5] Chain-of-Thought Prompting Elicits Reasoning in Large Language Models. NeurIPS 2022
>
> [6]  Language models of code are few-shot commonsense learners. EMNLP 2022
>
> [7] Least-to-Most Prompting Enables Complex Reasoning in Large Language Models. ICLR 2023
>
> [8] Self-Consistency Improves Chain of Thought Reasoning in Language Models. ICLR 2023
>
> [9] LESS: Selecting influential data for targeted instruction tuning. ICML 2024
>
> [10] Data selection for language models via importance resampling. NeurIPS 2023

---

> ### Author Response · Authors · 2024-11-20
> **Response to Reviewer k1j8 (2/2)**
>
> > Weakness 3: This method is constrained by the effectiveness of the LLM which generates code plannings for training.
>
> First, we want to clarify that the ability to **predict plans** given prompts is not bounded by the ability to **translate responses into plans". In other words, constructing code-form plans is an easier task. Our main experiments are all based on generating code-form plans using llama-3-8B, a moderate-scale open-source LM. Experiment results in Table 3 and Figure 4 already demonstrated the consistent and substantial improvements with such a cost-effective LM.
>
> In addition, we want to emphasize that the main contribution of our paper is to propose the CodePlan framework. After verifying the effectiveness of this idea using a moderate-scale open-source LM, we leave the exploration of using larger LMs (e.g. GPT-4) to construct code-form plans for future work. Interestingly, our preliminary experiment results in Appendix B.1 showed that using Llama-3-70B-Instruct to generate code-form plans even leads to slightly decreased performance.
>
>
> > Weakness 1: The font size in Table 1 is too small.
>
> We will enlarge the font size in Table 1 in the final version.

---

> ### Author Response · Authors · 2024-11-24
> **Looking forward to your comment**
>
> We have added further clarifications and experiments to address your reviews. Please feel free to let us know if you have any questions. Thanks a lot!!

---

> ### Comment · Reviewer_k1j8 · 2024-11-27
>
> I have read the author's response. Besides python-like planning, there are many other structured planning forms, such as markdown, YAML, json, html, etc. However, the author did not compare these other structured planning methods. Simply comparing with CoT is not sufficient. Please refer to: _Does Prompt Formatting Have Any Impact on LLM Performance_. Although the target tasks are different, this 4-page paper provides a more comprehensive comparison of LLM prompt formats. Therefore, I maintain my rating.

---

> > ### Author Response · Authors · 2024-11-28
> >
> > >  Besides python-like planning, there are many other structured planning forms, such as markdown, YAML, json, html.
> >
> > Thanks for your great question! Given the cost of training on Webinstruct and the limited time of the discussion phase, we conducted experiments on a moderately sized subset of MetaMath (12K examples). Specifically, we fine-tuned LLaMA-3-8B-instruct using different forms of plans generated by the model itself. The table below shows the accuracy of CodePlan and several representative baselines on GSM8K and MATH:
> >
> > |          | GSM8K    | MATH     |
> > | -------- | -------- | -------- |
> > | Base     | 78.5     | 29.8     |
> > | Vanilla  | 77.0     | 32.4     |
> > | JsonPlan | 75.8     | 35.2     |
> > | YamlPlan | 75.0     | 34.5     |
> > | MarkdownPlan | 74.4     | 35.2     |
> > | CodePlan | **80.5** | **36.8** |
> >
> > Code consistently outperforms other structured planning forms such as JSON, YAML and Markdown. On GSM8K, while other plan forms lead to performance degradation compared to the vanilla baseline (-2.6% - 1.2%), Code achieves a meaningful improvement (+3.5%). On a more challenging benchmark MATH, while introducing all plan forms leads to improvements, code still performs the best.
> >
> > Overall, while large-scale planning training alleviates the high variance in prompting as shown in [1], JSON, YAML, and Markdown all suffer from similar limitations in effectively capturing the structured high-level plans in diverse multi-step reasoning problems. We will incorporate these results in the revision.
> >
> > [1]  Does Prompt Formatting Have Any Impact on LLM Performance.

---

### Official Review · Reviewer_guz9 · 2024-11-03

**Soundness:** 2
**Presentation:** 3
**Contribution:** 2
**Rating:** 5
**Confidence:** 4

**Summary:**

This paper proposes incorporating pseudocode-based reasoning into the model's response process, which can effectively improve model performance. To obtain these pseudocode reasoning processes, the authors suggest using a large model to expand the responses directly on the WebInstruct dataset. Through extensive experiments, the authors demonstrate that training the model on the expanded dataset significantly enhances its performance.

**Strengths:**

1. The experiments are extensive.
2. A mathematical understanding of the underlying principles was developed, making the entire framework more solid.

**Weaknesses:**

1.  If the goal is to prove the value of the code-form plan, I think there are two aspects to consider:
1.1 If the goal is to demonstrate the benefit of this reasoning format to the model, then it would be appropriate to compare it with Chain-of-Thought (CoT) reasoning. Showing that a model performs better when using its code-based planning compared to traditional CoT results would more directly establish the superiority of the CodePlan format.
1.2 If the aim is to showcase the advantages of the code-form plan format and its ability to be automatically generated at scale with strong results, then the baseline of natural language enhancement should be more rigorously established. However, in the paper, the adaptation of natural language reasoning prompt (in Table 7) seems only slightly modified from the code-plan one, making the comparison less convincing.
[More comments on this prompt issue, natural language tends to be more redundant than code. Imposing a requirement like “The logic should be high-level, instead of replicating low-level details in the response” inherently makes it more challenging for the model. Additionally, as shown in the results of Table 4, modeling \( \log P(Z|X) \) is challenging, which partly suggests that the difficulty lies in generating this logic itself. This doesn't necessarily indicate that CodePlan is more effective; rather, it suggests that generating code-based plans may be easier because the task is easier. This easiness could be due to the prompts being optimized for generating code rather than for producing natural language explanations.]


2. The baseline results are different from Results in MAmmoTH2 (https://arxiv.org/pdf/2405.03548) . In Table 2 from MAmmoTH2, Mistral with WebInstruct, achieves 68.4/36.7 in GSM8k\MATH. While in this paper, Table 3 shows 54.1/31.5, which are much lower than reported.


3. Since the data generation process used LLaMA3 8B, the Plan-and-Solve (PS) Prompting baseline should also use LLaMA3 8B to generate the natural language plan prompt.

**Questions:**

1. The paper used LLaMA3 8B for generation—is this a type of distillation? It would be necessary to report LLaMA3 8B's performance on the evaluation datasets directly. Additionally, given that the size of LLaMA3 8B is quite similar to the model discussed in the paper, why not simply use LLaMA3 8B directly?

---

> ### Author Response · Authors · 2024-11-20
> **Response to Reviewer guz9**
>
> > Weakness 1(1): Regarding the goal of our paper and the comparison with Chain-of-thought (CoT) training
>
> CoT training is exactly our vanilla training baseline. Please see General Response 1.
> Moreover, we would like to emphasize that our primary goal is not to investigate a format but rather to investigate a fundamental approach that supervises LMs with explicit, structured (code-form), scalable planning signals.
>
>
> > Weakness 1(2): Regarding prompt engineering for natural language plan generation
>
>
> Our primary goal in this paper is to demonstrate the effectiveness of the proposed CodePlan framework. We intentionally avoided extensive prompt engineering for both code-form and natural language plans during data construction because:We aimed to minimize bias when constructing plans for diverse, large-scale datasets, rather than focusing on specific downstream tasks.
> The measurement of plan quality remains an open challenge in the field. As our results in Table 4 indicate, more verbose plans don't necessarily lead to improved performance, as they can also introduce greater challenges in plan generation.
> In particular, we believe “planning” should by definition be high-level and not replicate low-level details. Thus we apply this instruction for generating both code-form and natural language plans.
>
> In addition, It's worth noting that our approach to plan generation prompts is comparable to other recent works in the field. For instance, the Plan-and-solve method [1] uses similarly straightforward prompts. Here are two examples from their paper:
> General Prompt: “Let’s first understand the problem and devise a plan to solve the problem.”
> Task-Specific Prompt (GSM8K): “Let’s first understand the problem, extract relevant variables and their corresponding numerals, and make a complete plan”.
> These two plan generation prompts only lead to a difference of 1% in GSM8K accuracy in their paper.
>
> To further validate the performance of our prompt for natural language plan construction, we conduct a comparison study to Plan-and-Solve[1]. Specifically, given the same set of few-shot examples, we generate natural language plans using the general prompt and task-specific prompt from [1], as well as our prompt, and compare the few-shot accuracy on GSM8K using Mistral-7B as the base model. As shown in the Table below, our prompt leads to an even slightly better performance than [1], indicating its effectiveness.
>
> |                                       | Few-shot Accuracy |
> | ------------------------------------- | ----------------- |
> | Plan-and-solve (General Prompt)       | 44.9              |
> | Plan-and-solve (Task-specific Prompt) | 45.1              |
> | Plan-and-solve (Our Prompt) | 45.5              |
>
>
>
> > Q1: Regarding the experiments using llama-3-8B.
>
>
> In our original experiments, we aim to study the effectiveness of CodePlan under different model scales. We used Llama-2 models because there is no official ~13B model in the Llama-3 series. In General Response 4, we clarified that CodePlan is not a type of distillation. In General Response 5, we conducted additional experiments using Llama-3-8B to provide a more comprehensive evaluation.
>
>
> > Weakness 2: Regarding using llama-3-8B to generate natural language plans.
>
> We would like to clarify that we have indeed implemented and thoroughly discussed this baseline
> - For the Plan-and-solve Prompting results in Table 3, we used llama3-8b-generated natural language plans in the few-shot prompts.
> - We further used llama-3-8b-generated natural language plans to augment our training data and replicated the training pipeline of CodePlan. Results are shown in Finding 3 of Section 3.6 (Figure 4 and Table 4). Our experimental results demonstrate that CodePlan consistently and substantially outperforms its natural language counterpart. Please see General Response 2 for more discussions.
>
>
> > Weakness 2: Regarding the difference in baseline results compared to the MAmmoTh2 paper
>
> The difference in performance is mainly attributed to the dataset size used in the respective studies. MAmmoTH2 utilized its full dataset, which comprises 10M examples. In contrast, our study was based on the publicly released subset of 2M examples. We will highlight this distinction in our revision.
>
> Regarding the potential of CodePlan, our training curves (Figure 3) suggest that the performance gap between CodePlan and vanilla training may widen with increased training data. We are eager to evaluate CodePlan using the full 10M example dataset once it becomes accessible.

---

> > ### Comment · Reviewer_guz9 · 2024-11-26
> > **Is there a result from a model self-generated CodePlan to enhance itself?**
> >
> > The first two reasons in the response to the question "Is CodePlan a type of distillation?" do not adequately explain that it is not distillation, as using data generated by llama3-8b as the target is an implicit form of distillation. Is there an example where CodePlan generated by the same model is used to further train and enhance that model? (For example, LLaMA3 can benefit from its own codeplan.)

---

> ### Author Response · Authors · 2024-11-24
> **Looking forward to your comment**
>
> We have added further clarifications and experiments to address your reviews. Please feel free to let us know if you have any questions. Thanks a lot!!

---

> > ### Comment · Reviewer_guz9 · 2024-11-26
> > **Why not using LLaMA3 8B directly?**
> >
> > I am not sure whether this question "Additionally, given that the size of LLaMA3 8B is quite similar to the model discussed in the paper, why not simply use LLaMA3 8B directly?" has been addressed in the response. Can you make it clear in one response?

---

> > > ### Author Response · Authors · 2024-11-27
> > >
> > > In our original experiments, we aim to study the effectiveness of CodePlan under different model scales. We used Llama-2 models because there is no official ~13B model in the Llama-3 series.
> > > We also presented llama-3 results in general response (2/2).

---

> ### Author Response · Authors · 2024-11-27
>
> Given the cost of training on Webinstruct and the limited time of the discussion phase, we conducted experiments on a moderately sized subset of MetaMath (12K examples). Specifically, we fine-tuned LLaMA-3-8B-instruct using code-form plans generated by the model itself. The table blow shows the accuracy of CodePlan and several representative baselines on GSM8K and MATH:
>
> |          | GSM8K | MATH |
> | -------- | ----- | ---- |
> | Base     | 78.5  | 29.8 |
> | Vanilla  | 77.0  | 32.4 |
> | CodePlan | 80.4  | 36.8 |
>
> These results demonstrate that CodePlan maintains its effectiveness even when utilizing self-generated plans, showing consistent improvements over both the base model and vanilla fine-tuning. On GSM8K, while vanilla training showed a slight performance degradation (-1.5%) compared to the base model, CodePlan achieved a meaningful improvement of 1.9% absolute accuracy over the base model. The gains are even more pronounced on the more challenging MATH dataset, with a 7.0% absolute improvement over the base model and a 4.4% improvement over vanilla fine-tuning.
>
> These findings suggest that the structured nature of code-form plans provides valuable guidance for reasoning, even when the plans are generated by the model itself. This self-improvement capability aligns with our broader vision of CodePlan as a scalable paradigm for enhancing LLMs' reasoning abilities. In the camera-ready version, we will include these additional experiments and insights in the main paper.

---

> ### Comment · Reviewer_guz9 · 2024-11-27
>
> Ok, I will raise my score to 5. In the additional experiment, is the CodePlan generated by LLaMA3-8b-instruct or LLaMA3-8b-base?

---

> ### Author Response · Authors · 2024-11-27
>
> In the additional experiment in General Response (2/2), we fine-tuned llama-3-8b on WebInstruct using the code-form plans generated by llama-3-8b-instruct.
>
> If you are interested in all self-training results where the model is trained on its own generated CodePlan, we now have the results for both [instruct model](https://openreview.net/forum?id=dCPF1wlqj8&noteId=B8awS3E2Bw) (using CodePlan, where the plan is generated by conditioning on golden reasoning steps) and [base model](https://openreview.net/forum?id=dCPF1wlqj8&noteId=FAO1LbRbZI) (using STaR + CodePlan, where the plan is generated by only using the question). In both setups, CodePlan achieves substantial improvements.
>
> We would be happy to add further experiments and clarifiactions to improve the understanding of our work! Thanks for your great questions!

---

### Official Review · Reviewer_5Vzm · 2024-11-03

**Soundness:** 3
**Presentation:** 2
**Contribution:** 3
**Rating:** 6
**Confidence:** 4

**Summary:**

This paper proposes an approach to improve LLM performance on varied reasoning tasks including mathematical and symbolic reasoning and multi-hop QA. The approach fine-tunes an LLM to predict Python pseudo-code "plans" from a natural language query, and to then condition on this pseudo-code to generate the answer to the query (the code is not executed). To obtain the pseudo-code plans for training, an LLM is prompted using queries and ground-truth solutions on 2M query/response pairs from WebInstruct. The resulting query, pseudo-code, solution tuples are then used to fine-tune the LLM. The paper evaluates on three different base LMs, and a variety of datasets including GSM8K, MATH, SVAMP, and HotpotQA, finding substantial improvements over direct prediction of answers and an variant of the method that uses natural language plans rather than pseudo-code.

**Strengths:**

S1) The paper's experimentation was overall thorough:
- It applied its training procedure to three different open source models, and evaluated them across a range of datasets covering various types of reasoning where code might potentially help.
- The analysis experiments, in particular Finding 3 that used natural language plans, were a helpful step toward understanding the approach (but see below for a few questions/points on this).

S2) The results seemed strong, finding that the approach obtains large improvements over vanilla (non-planning) training and using natural language plans.

S3) I found the paper well-motivated, given the findings of past papers that training on code may improve reasoning abilities [1,2,3], and that models can be trained to reason (in natural language) using approaches like STaR.

[1] The Magic of IF: Investigating Causal Reasoning Abilities in Large Language Models of Code. Liu et al. 2023
[2] To Code, or Not To Code? Exploring Impact of Code in Pre-training. Aryabumi et al. 2024.
[3] If LLM Is the Wizard, Then Code Is the Wand. Yang et al. 2024.

**Weaknesses:**

W1) While the paper took steps toward explaining it (via the NLL experiments in Table 4), I still found it a bit underanalyzed *why* using pseudocode plans works better than natural language plans. It would be useful to also investigate whether the pseudocode plans are also more accurate or otherwise higher in quality semantically than natural language plans, perhaps doing some human evaluation on a small number of samples.

W2) The experimental setup left a few stones unturned which would be natural to investigate:
- The model used to produce the code plans is Llama3-8B-Instruct, which is different (and potentially stronger) than any of the models fine-tuned on the plans (Mistral-7B, Llama-2-7B and 13B), so there's an element of model distillation in the paper. It would be helpful to evaluate whether a model could bootstrap itself as in the STaR paper, or at least to add a note on this.
- The evaluation could be more rigorous in places, see questions and suggestions below.

W3) While I think this could definitely be fixed for a camera-ready version of the paper, the writing of the paper could be a bit improved:
- Make it clearer that in no cases (if I understand right) are the code plans being executed -- they are just conditioned on by the LLM. It would help to call them "pseudo-code" consistently throughout the paper (rather than just in a few places as it is currently).
- Make the comparison clearer to STaR (as the method is similar) and Quiet-STaR (as their results are reported). The "planning in natural language" baseline in Finding 3 has some similarities to STaR (training the model to produce latent natural language plans) but also some key differences (all annotations are produced using query and answer; a different model is used to annotate than is being trained).
- I didn't feel that Table 1, with the comparison between CodePlan and other methods, added much to the paper as many of the attributes are unexplained.
- Clarify some details -- see questions below.

**Questions:**

Q1) Did the authors train Quiet-STaR to get the numbers in Table 3, or the results are from the model from the original paper? I found it strange that Quiet-STaR is worse on nearly every metric and dataset compared to the Mistral-7B version, is there any explanation for this?

Q2) It wasn't clear to me how CodeReason works: is the code actually executed, or does the prompt just encourage the model to produce executable code (but otherwise it works the same as CodePlan)?

Q3) "models are instructed to generate CoT-style responses". Are these CoT also included in the training data (if not, seems there is a mismatch between training and inference)?

Q4) do the few-shot examples also show Code Plan examples?

Q5) For the case studies in appendix B.3, are these code plans generated by the model itself (from only the query), or are the annotated code plans used in training (from the query and answer)?

*Minor Questions / Suggestions (no need to respond in the author response)*

- "delicate balance" part in 2.1 is a bit vague

- Figure 1: the digit example makes sense, but the renewable energy didn't to me. Where do the digits come from? There is some fact-based reasoning in the "response" text for *why* solar energy has limited accessibility, and it's not clear to me why it would be easier for the model to predict the numeric scores (needed to produce the code plan) rather than the facts. Is the main advantage of code here just to make sure all energy sources are mentioned, without repetition, and contrasted with each other?

- Many of the attributes in Table 1 are subjective, e.g. what makes this work efficient but QuietSTaR not? Why is CoT interpretable but QuietSTaR is not?

- Is "NLL" the average token-level NLL?

- "code is far more prevalent in pre-training corpora than natural language plans". This is an interesting claim, but needs evidence. "natural language plan" is probably harder to define and so also probably hard to quantify?

- Why is a 13B model necessary (the reason given in footnote 1 for why Llama-3 was not used)? Mistral-7B was used, and there is a ~7B Llama-3 model.

- "we synthesize 5K examples based on official implementation" implementation of what?

- The Kingma 2013 citation should probably be Kingma and Welling.

- There are a number of misspellings throughout the paper.

---

> ### Author Response · Authors · 2024-11-20
> **Response to Reviewer 5Vzm**
>
> > Weakness 1: Regarding investigating the quality of code-form plans.
>
> Thanks for the suggestion! However, measuring the quality of a training example (i.e., its influence on the final model performance) is still an open problem ([1][2]). For example, as indicated by our results in Table 4, more verbose natural language plans do not lead to higher performance compared to concise code-form plans. This may be because verbose plans, while potentially more detailed, also pose a greater challenge for plan generation. Similarly, an executable code plan does not necessarily lead to higher performance than its non-executable counterpart.
>
> After demonstrating the effectiveness of code-form planning in this paper, we will continue exploring measuring and improving the quality of code plans in future work. This is similar to how the initial demonstration of CoT prompting [3]  laid the groundwork for subsequent investigations into advanced CoT realizations [4][5].
>
> > Weakness 2 (1) & 3 (2): Regarding the comparison with STaR and Quiet-STaR, and the performance of CodePlan using llama-3-8b.
>
> Please see General Response 3 for the comparison between CodePlan and STaR/Quiet-STaR. In General Response 4, we clarify that CodePlan is not a type of distillation. Please see General Response 5 for results using llama-3-8b.
>
> > Q3: Regarding whether the response includes CoT during training.
>
> Yes, the responses in our training dataset consist of free-form text reasoning steps, not merely final answer spans. Please see General Response 1 for more details.
>
>
> > Q5: Regarding code plans in the case studies
>
> The code plans presented in these cases are generated by the model itself, using only the query as input. These are not the annotated code plans used during training.
>
> We deliberately chose this approach to demonstrate the model's ability to generate code-form plans for tasks it hasn't been specifically trained on. This aligns with our goal of verifying the generalization of CodePlan.
>
>
>
> > Q1: Regarding the results of Quiet-STaR.
>
> We trained Quiet-STaR ourselves using the official code on the same dataset as CodePlan to ensure a fair comparison. We used the default hyperparameters but did not observe substantial improvements. We have not found any other published studies replicating the results of Quiet-STaR, making it challenging for us to draw a definitive explanation for this observation.
>
>
>
> > Weakness 3 (1): Regarding calling the plans “pseudo-code”
>
> Thanks for your advice! We will further emphasize these pseudocode plans are not executed.
>
>
> > Q2: Regarding how CodeReason works.
>
> CodeReason would directly execute the generated programs, and use the execution results as the final answer. Instead, CodePlan would first generate pseudocode, and generate natural language reasoning paths to produce the final answer.
>
>
> > Q4: Regarding whether the few-shot examples also include code-form plans.
>
> Yes.
>
>
> [1] LESS: Selecting influential data for targeted instruction tuning. ICML 2024
>
> [2] Data selection for language models via importance resampling. NeurIPS 2023
>
> [3] Chain-of-Thought Prompting Elicits Reasoning in Large Language Models. NeurIPS 2022
>
> [4] Least-to-Most Prompting Enables Complex Reasoning in Large Language Models. ICLR 2023
>
> [5] Self-Consistency Improves Chain of Thought Reasoning in Language Models. ICLR 2023

---

> > ### Comment · Reviewer_5Vzm · 2024-11-28
> > **Thanks for the response**
> >
> > Thanks to the authors for the response, and apologies for the delay here! I still feel positively about the paper.

---

> ### Author Response · Authors · 2024-11-24
> **Looking forward to your comment**
>
> We have added further clarifications and experiments to address your reviews. Please feel free to let us know if you have any questions. Thanks a lot!!

---

### Official Review · Reviewer_Jy2Q · 2024-11-04

**Soundness:** 2
**Presentation:** 3
**Contribution:** 2
**Rating:** 3
**Confidence:** 4

**Summary:**

This paper describes a way to improve LLM reasoning ability by training it to plan in the form of code. Specifically, it first prompts LLM to generate code form plan given existing prompts and responses. Then, it finetunes LLM to generate plan given prompt, and generate response given plan and prompt. Experimental results show performance improvements across over the baselines on multiple benchmarks.

**Strengths:**

- The paper is well written
- The experiments are extensive, over multiple types of models and benchmarks

**Weaknesses:**

- The main novelty of this paper is that it uses code form rationale/planning instead of free text form. Besides code form, the specific way of training LLM to predict a prompted rationale is already proposed by the paper STAR https://arxiv.org/pdf/2203.14465 (see the Rationalization section). This is not clear from the paper.
- Given that the main novelty is code form plan, its effectiveness over a free text form plan is not demonstrated from the experiments. The authors should include a comparison against STAR, and an ablation study on replacing code form plan with text form plan with the exact same pipeline.

**Questions:**

See weakness.

---

> ### Author Response · Authors · 2024-11-20
> **Response to Reviewer Jy2Q**
>
> > Weakness 1 & 2: Regarding the novelty of CodePlan and the comparison with natural language planning and STaR.
>
> We’d like to emphasize that the main novelty of our work lies in supervising LMs with explicit, structured (code-form), scalable planning signals, rather than merely proposing a format.
>
> We have indeed thoroughly compared code-form plans to natural language plans in Finding 3 of Section 3.6  (Figure 4 and Table 4). Please see General Response 2 for more details. Regarding the comparison with STaR, please see General Response 3.

---

> ### Author Response · Authors · 2024-11-24
> **Looking forward to your comment**
>
> We have added further clarifications and experiments to address your reviews. Please feel free to let us know if you have any questions. Thanks a lot!!

---

### Official Review · Reviewer_6LBD · 2024-11-04

**Soundness:** 3
**Presentation:** 3
**Contribution:** 3
**Rating:** 8
**Confidence:** 4

**Summary:**

This paper proposed to train LLMs to write code-form plans before performing reasoning on downstream tasks. The code-form plans are pseudocode written by Llama3-8B, and such augmentation is conducted on the 2M WebInstruct data (from Mammoth2 paper). They trained multiple models (mistral/llama2-7b/llama2-13b) on these data and observed large performance improvements over vanilla fine-tuning and training with language-form plans. Such performance gain is observed on 13 benchmarks including math reasoning, symbolic reasoning, instruction following, and multi-hop QA.

**Strengths:**

1. The authors proposed a novel idea (CodePlan): training LLMs to write code-based plans before conducting natural language reasoning. This idea utilizes the conciseness and structure of code to form high-level plans, and can benefit from LLMs' strong coding knowledge learned from pre-training.
2. The authors built a large-scale (2M) fine-tuning dataset with code-form plans based on WebInstruct. This resource could be valuable for future research.
3. The authors conducted solid experiments to prove CodePlan's supriority over vanilla fine-tuning and training with natural language plans. The benchmarks are tested out-of-distribution (no task-specific fine-tuning), which proves the generalization of the model.

**Weaknesses:**

I didn't find any critical weakness that would lead to rejecting the paper. However, the paper can be improved from the following perspectives:

1. The training data can be further improved. The current data is composed of two components: a pseudocode plan and the original natural language answer. I have the following concerns:

(1) The original answer is not further modified to better align and be more coherent with the pseudocode plan. This could limit the effectiveness of the training process.

(2) Two components seem separated and are not sufficiently integrated together as a coherent response, which does not look like a real human-style thinking procedure. As a result, models are still adapted to this specific response format instead of a universal reasoning strategy like CoT or the think-and-reason style of GPT-o1.

(3) The authors mentioned in the analyses that code-based reasoning are more effective than CodePlan on some tasks. So it may be worth exploring a combination of these two strategies. For example, there are papers that combined reasoning and code writing [1,2], which can be used to integrate with CodePlan for a more universal solution for model reasoning.

2. The authors did not conduct experiments on recent open-source models. The tested models (Mistral/llama-2) are released more than a year ago and are known for limited reasoning abilities. For example, fine-tuned llama-2 is still <50% on GSM8k while the more recent llama-3 is ~80%.
3. For presentation: the language-form planning baseline should be moved to the main table for a more comprehensive comparison, as I think this is an important baseline to show the effectiveness of CodePlan. Vanilla fine-tuning is a weak baseline because additional data are augmented into CodePlan's fine-tuning.

[1] MathCoder: Seamless Code Integration in LLMs for Enhanced Mathematical Reasoning\
[2] ToRA: A Tool-Integrated Reasoning Agent for Mathematical Problem Solving

**Questions:**

Most tables and figures in the paper are hard to recognize. I strongly suggest increasing the font size of these tables and figures, including Figure 2,3,4, and Table 3. For Table 3, do not separate the table into two parts. You don't need to report multiple metrics for multi-hop QA. Use abbreviations for dataset names.

---

> ### Author Response · Authors · 2024-11-20
> **Response to Reviewer 6LBD**
>
> > Weakness 2: Regarding the experiments on recent open-source models.
>
> Please see General Response 3.
>
>
> > Weakness 3: Regarding the presentation of the natural language planning and vanilla training baselines
>
> We will further highlight the comparison between planning in code and natural language in our revision. In addition, we argue that vanilla training is not a weak baseline, please see General Response 1 for more explanations.
>
>
> > Weakness 1 (2): Regarding the CodePlan format.
>
> We would like to respectfully offer a different perspective on the integration and universality of our approach:
>
> - **Human-like reasoning**: “Plan-and-reason” actually mirrors human cognitive processes for complex tasks. This is evident in various domains including:
>     - Software development: Programmers typically design modular structures and overall logic before writing detailed code line by line [1].
>      - Writing: Authors frequently create outlines before writing detailed narratives [2].
> - **Universality of deliberate planning**: We argue that plan-and-reason is as universal an approach as Chain-of-Thought (CoT) or GPT-4's think-and-reason style. Recent research has demonstrated the efficacy of deliberate planning across diverse domains, including mathematics [3], question-answering [4], and agentic tasks [5]. While these studies primarily focus on *natural language planning* through *task-specific* prompts or curated fine-tuning data, CodePlan innovates by leveraging *code-based planning* learned from *massive, wide-ranging* text corpora.
> - **Beyond formatting**: CodePlan is not merely a response format but a fundamental approach that *supervises LMs with explicit, structured (code-form), and scalable planning signals* to enhance their reasoning capabilities. Our experiment results highlight its superiority in reasoning performance (Table 3), data efficiency (Figure 3), and transferability (Table 15).
> In summary, we believe that CodePlan offers a novel and powerful approach to structured reasoning in LLMs.
>
>
>
> > Weakness 1 (1): Regarding the coherence between the plan and response.
>
> In our current implementation, the original answer is not further refined to align with the generated pseudocode plan. This is because our primary focus in this work is on demonstrating the effectiveness of code-form plans in guiding the reasoning process.
>
> We acknowledge that further refining final answers to align with the generated plan could potentially further enhance model performance. After establishing the effectiveness of CodePlan, we would explore iterative refinement techniques in future work.
>
>
>
> > Weakness 1 (3): Regarding the combination of CodePlan and CodeReason.
>
> We agree this is an interesting direction. However, this is beyond the scope of our paper, so we leave it for future work.
>
> [1] Parsel : Algorithmic Reasoning with Language Models by Composing Decompositions. NeurIPS 2023.
>
> [2] DOC: Improving Long Story Coherence With Detailed Outline Control. ACL 2023
>
> [3] Plan-and-solve prompting: Improving zero-shot chain-of-thought reasoning by large language models. ACL2023
>
> [4] Amor: A recipe for building adaptable modular knowledge agents through process feedback. NeurIPS 2024
>
> [5] AutoGPT: Build, Deploy, and Run AI Agents. https://github.com/Significant-Gravitas/AutoGPT

---

> > ### Comment · Reviewer_6LBD · 2024-11-20
> > **Response to Authors**
> >
> > Regarding weakness 2: Thank you for your additional experiments on Llama-3. They look good.
> > Regarding weakness 1: Thank you for your clarifications, although I did not intend to question the novelty of your paper. I am just saying that this approach can be further refined to make it more generalizable and universal (e.g., to apply it in real production). For this submission itself, I think it contains enough contribution for a research paper.

---

### Author Response · Authors · 2024-11-20
**General Response (2/2)**

> 3. What’s the difference between CodePlan and STaR or Quiet-STaR?

We clarify and expand on the distinctions between these approaches in the table below:

| **Method**           | **Data Souce**  | **High-Level Plan**  | **Low-Level Reasoning Steps** | **Final Answer Span Required** |
| -------------------------------- | -------------------------- | ---------------------------- | ----------------------------- | ------------------------------ |
| **Vanilla (i.e., CoT Training)** | Web-crawled Unlabeled Data | N/A     | Golden    | ×       |
| **STaR**                         | Downstream Labeled Data    | N/A       | Search Through Bootstrapping  | ✓     |
| **Quiet-STaR**                   | Web-crawled Unlabeled Data | N/A    | Search Through REINFORCE      | ×   |
| **Natural Language Planning**    | Web-crawled Unlabeled Data | Machine Annotation           | Golden    | ×   |
| **CodePlan**                     | Web-crawled Unlabeled Data | Machine Annotation           | Golden                        | ×                              |
| **STaR + CodePlan**              | Downstream Labeled Data    | Search Through Bootstrapping | Search Through Bootstrapping  | ✓                              |

Our experimental results in Table 3 demonstrate that CodePlan consistently outperforms vanilla training (i.e., CoT training) and Quiet-STaR across various reasoning tasks. This suggests that the abstraction provided by code-form plans offers tangible benefits in reasoning performance.

Additionally, we conducted an additional experiment in response to Reviewer 5Vzm’s interest in studying bootstrapped code-form plans under the STaR framework (i.e., STaR+CodePlan in the table above). Considering that STaR requires final answer spans for checking the correctness of the bootstrapped reasoning steps, we adopt a subset of MetaMath [3] consisting of 20K examples with questions and final answer spans. Using the official STaR implementation, we evaluated model performance on the GSM8K benchmark. The results are as follows:

| **Method**                       | **Accuracy** |
| -------------------------------- | ------------ |
| **Vanilla (i.e., CoT Training)** | 54.4         |
| **STaR**                         | 46.8         |
| **CodePlan**                     | **61.2**     |
| **STaR + CodePlan**              | 51.7         |

These results indicate that STaR+CodePlan yields better reasoning performance (51.7 vs. 46.8) than standard STaR. However, this approach still underperforms the Vanilla training baseline which is fine-tuned on golden reasoning steps. We will add the results in our revision.

> 4. Is CodePlan a type of distillation?

We want to emphasize that CodePlan is not a type of distillation. Here are some key points to consider:

- While we used llama-3-8b to translate golden responses into code-form plans for training data construction, our model is fine-tuned to directly predict plans given only the prompt. Therefore, its reasoning ability is not bounded by llama-3-8b.
- We did not conduct fine-tuning on downstream tasks
- As shown in the self-bootstrapping experiments (CodePlan+STaR) in General Response 3, CodePlan is superior to standard STaR. The results further support that CodePlan is not merely distilling knowledge from another model, but is actively enhancing reasoning capabilities.

> 5. How does CodePlan perform on recent LMs such as Llama-3?

Since llama-3-8b-instruct might already be trained on downstream datasets ([6]), we conducted experiments based on llama-3-8b-base. As shown in the table below, CodePlan still consistently and substantially outperforms both vanilla fine-tuning and natural language planning. We will add the experiment results of llama-3 on all evaluation benchmarks in the final revision.


|                       | Mathematical Reasoning | Symbolic Reasoning | Multi-hop QA |
| --------------------- | ---------------------- | ------------------ | ------------ |
| Base                  | 39.1                   | 57.6               | 27.2         |
| Vanilla               | 45.2                   | 62.2               | 28.1         |
| Natural Language Plan | 46.8                   | 62.7               | 31.2         |
| CodePlan              | **49.7**               | **71.3**           | **33.5**     |


[1] Scaling instruction-finetuned language models. JMLR 2024

[2] Wizardmath: Empowering mathematical reasoning for large language models via reinforced evol-instruct. arXiv 2024

[3] MetaMath: Bootstrap Your Own Mathematical Questions for Large Language Models. ICLR 2024

[4] MAmmoTH2: Scaling Instructions from the Web. arXiv 2024

[5] Instruction Pre-Training: Language Models are Supervised Multitask Learners. EMNLP 2024

[6] VarBench: Robust Language Model Benchmarking Through Dynamic Variable Perturbation. EMNLP 2024

---

### Author Response · Authors · 2024-11-20
**General Response (1/2)**

We thank all reviewers for their insightful feedback! Here we address the shared questions from reviewers:


> 1. How does CodePlan compare to traditional CoT training? Does vanilla training involve CoT reasoning paths? Is vanilla training a weak baseline?

We would like to emphasize that the “Vanilla training” baseline exactly refers to the traditional CoT training, since the response in our training dataset consists of free-form text reasoning steps, not merely the final answer span. This baseline is widely adopted to build recent LMs ([1][2]). Therefore, we regard it as a strong baseline to reveal the benefit of explicitly planning. Our results in Table 3 show that CodePlan achieves a 25.1% relative improvement compared to Vanilla training, averaged across 13 challenging multi-step reasoning benchmarks.

> 2. How does CodePlan compare to planning in natural language? Why does CodePlan outperform its natural language counterparts?

For clarity, here's an example of natural language planning and code-form planning for a MATH question:

```
Question (from MATH):
Find the least common multiple of $6!$ and $(4!)^2.$

Natural Language Plan:
1. Factorize the numbers: Break down $6!$ and $(4!)^2$ into their prime factors.
2. Identify common factors: Find the common prime factors between the two factorizations.
3. Take the highest power: For each common prime factor, take the highest power that appears in either factorization.
4. Multiply the highest powers: Multiply the highest powers of each common prime factor to find the least common multiple.

Code-form Plan:
def lcm_factorial_powers(fact1, fact2):
    fact1_prime = prime_factorize_factorial(fact1)
    fact2_prime = prime_factorize_factorial(fact2)
    lcm_primes = combine_highest_powers(fact1_prime, fact2_prime)
    lcm_value = compute_product(lcm_primes)
    return lcm_value

Response:
6!=2^4*3^2*5 and 4!^2=(2^3*3^1)^2=2^6*3^2
The least common factors with the highest power is 2^6, 3^2, 5^1
Mulitply the highest powers we have 2^6\cdot3^2\cdot5^1=2880

The answer is $2^6\cdot3^4\cdot5^2$
```


Figure 4 and Table 4 in Finding 3 of Section 3.6 show the comparison results between CodePlan and a natural language plan.
- Firstly, CodePlan consistently outperforms natural language planning across various tasks:
    - Mathematical reasoning: 4% relative improvement
    - Symbolic reasoning: 27.2% relative improvement
    - Instruction following: 6.5% relative improvement
    - Decision-making tasks: 27.5% relative improvement
- Secondly, as shown in Table 4, while natural language planning significantly reduces losses for surface realization, it introduces a substantial challenge in the planning stage compared to CodePlan.

We believe CodePlan is superior for two reasons: (1) the structured and precise nature of code allows for an easier-to-learn representation of complex control flows inherent to sophisticated reasoning, and (2) LLM can better leverage their pre-training knowledge, as code is far more prevalent in pre-training corpora than natural language plans.

---

### Meta-Review · Area_Chair_YMff · 2024-12-23

**Metareview:**

Accept conditioning on including llama3-8B results in the main table.

This paper proposed to train LLMs to write code-form plans before performing reasoning on downstream tasks. The code-form plans are pseudocode written by Llama3-8B, and such augmentation is conducted on the 2M WebInstruct data (from Mammoth2 paper). They trained multiple models (mistral/llama2-7b/llama2-13b/llama-3) on these data and observed large performance improvements over vanilla fine-tuning and training with language-form plans. Such performance gain is observed on 13 benchmarks including math reasoning, symbolic reasoning, instruction following, and multi-hop QA.

Overall reviewers liked the breadth of the results (on Math, code but also QA) and the strength of the results under their experimental setting of training base models using llama-3-instruct generated data. Reviewers were slightly unsure how much ``distillation" there is and questioned comparisons to some baselines (e.g. STAR).

The experiment setup and whether there is distillation (teacher model, instead of logits) is a major issue. In the original setup, the authors used llama3-8B-instruct to generate data and then trained llama2-7B/13B and mistral. Reviewers and the AC did not find their original justification that llama3 did not have a 13B model to be a convincing reasons for exclusions. A key counter-argument from 6LBD is that their method is still better than using NL plans under the same setting. With that, I consider this issue to be mostly addressed but the original experiment setup left 3 reviewers and the AC questioning, so the authors are strongly encouraged to improve it in the final version.

I consider the 2 reject reviews here. Jy2Q: raised points on novelty comparing to STAR, this is fully addressed. k1j8: several points were not specific to this work (more broadly consider plan and logic-based approaches) and are too vague to be addressed. I down-weighted this review a bit because these points can be applied to almost all works in this area.

**Additional Comments On Reviewer Discussion:**

a bit of debates of the experimental setting, 6LBD helped convince the AC that the setup is okay. Thanks.

---

### Decision · Program_Chairs · 2025-01-22

Accept (Poster)